# ROBUSTNESS SIGNATURES IN CLIP MODELS

<CHANGES MADE DURING REBUTTAL IN MAGENTA>

## ABSTRACT

What distinguishes robust models from non-robust ones? This question has gained traction with the appearance of large-scale multimodal models, such as CLIP. These models have demonstrated unprecedented robustness with respect to natural distribution shifts. While it has been shown that such differences in robustness can be traced back to differences in training data, so far it is not known what that translates to in terms of what the model has learned. In this work, we bridge this gap by probing the representation spaces of 16 robust CLIP models with various backbones (ResNets and ViTs) and pretraining sets (OpenAI, LAION-400M, LAION-2B, YFCC15M, CC12M and DataComp). We find two signatures of robustness in the representation spaces of these models: (1) Robust models exhibit outlier features characterized by their activations, with some being several orders of magnitude above average. These outlier features induce privileged directions in the model's representation space; (2) Robust models encode substantially more concepts in their representation space. While this superposition of concepts allows robust models to store much information, it also results in highly polysemantic features, which makes their interpretation challenging. We also validate our findings on other robust multimodal models beyond the CLIP-family, namely on CoCa models.

## 1 INTRODUCTION

Large pretrained multimodal models, such as CLIP (Radford et al., 2021), have demonstrated unprecedented robustness to a variety of natural distribution shifts[1]. In particular, when used as zero-shot image classifiers, their performance on ImageNet (Deng et al., 2009) translates remarkably well to performances on natural shifts of ImageNet, such as, e.g., ImageNet-V2 (Recht et al., 2019). This led to many works analyzing *what actually causes this remarkable robustness of CLIP*, with Fang et al. (2022) establishing that the root cause of CLIP's robustness lies in the quality and diversity of data it was pretrained on.

While the cause of CLIP's robustness is known, we set out to establish how exactly robustness manifests itself in features learned by the model. Finding feature patterns that only appear in robust models is the first step towards a better understanding of the emergence of robustness. This understanding is key to diagnosing robustness in times when knowledge about the (pre)training distribution and/or the distribution shifts cannot be assumed.

To find robustness patterns in robust CLIP models, we leverage the various models provided by Ilharco et al. (2021) in the OpenCLIP repository. We analyze the *visual features* in these models by probing their last layer activation vectors with quantitative interpretability tools, such as kurtosis analysis of activation vectors (Elhage et al., 2023), singular value decomposition (SVD) of weight matrices and concept probing of the representation space (Bau et al., 2017). Through this analysis, we distill a set of distinctive characteristics of robust CLIP model features, which constitute the core contribution of our work. Interestingly, we find these signatures to also hold for other robust models that are not CLIP models, such as the multimodal CoCa (Yu et al., 2022) models or the pure vision NoCLIP (Fang et al., 2022) models (Appendix C).

---

[1]Similar to Radford et al. (2021), we use CLIP as a name for the general training technique of unsupervised language-vision pretraining, not only for the specific models obtained by OpenAI.

**Our contributions. (1)** In Section 3, we show that robust CLIP models all have *outlier features*. These features stand out as their activation is typically several orders of magnitude above the average activation of features in the same layer. Through an SVD analysis, we show that these outlier features are propagated to the logits of downstream classifiers, which results in what we call *privileged directions* that are crucial to model predictions. **(2)** In Section 4, we show that robust models all encode a high number of unique concepts in their features. As a consequence, the features of robust models are highly *polysemantic*, which means that they superpose a large set of concepts. **(3)** Crucially, we show that the predominance of outlier features, privileged directions and a high number of encoded concepts are signatures that distinguish robust models from their non-robust counterparts. We discuss the potential of using these signatures to diagnose robustness of trained models without the need to run inference on shifted distributions.

## 2 MEASURING ROBUSTNESS

In this section, we define *Effective Robustness* (ER), which is the type of robustness we study in this paper. We then measure the ER for the set of models extracted from OpenCLIP. While most of the results in this section are disseminated in the literature, we believe it is useful to reproduce them and present them in a unified way.

**Context.** ER has emerged as a natural metric to measure how the performance of a model on a reference distribution (in-distribution) generalizes to natural shifts of this distribution (Fang et al., 2022). In this work, we focus on ImageNet (Deng et al., 2009) and its natural distribution shifts considered in Fang et al. (2022), namely ImageNet-V2 (Recht et al., 2019), ImageNet-R (Hendrycks et al., 2021a), ImageNet-Sketch (Wang et al., 2019), ObjectNet (Barbu et al., 2019) and ImageNet-A (Hendrycks et al., 2021b). When plotting the in-distribution accuracy (X-axis, *logit scaling*) against the average shifted-distribution accuracy (Y-axis, *logit scaling*) of various architectures trained on ImageNet, Taori et al. (2020) found that most of the existing models lie on the same line. They also found that models trained with *substantially* more data lie above this line, showing a desirable gain in shifted-distribution accuracy for a fixed in-distribution accuracy. They coined this vertical lift above the line as *Effective Robustness*.

**Computing ER.** To quantify ER, following Taori et al. (2020), one gathers the ImageNet test accuracy $\text{ACC}(I)$ and the average accuracy over the ImageNet shifts $\text{ACC}(S)$ of a set of reference models trained on ImageNet and fits a linear model on this pool of accuracies to map $\text{logit}[\text{ACC}(I)]$ to $\text{logit}[\text{ACC}(S)]$, with the logit function $\text{logit} : [0, 1] \to \mathbb{R}$ defined as $x \mapsto \ln(x) - \ln(1 - x)$. The resulting line can be used to predict what (logit) accuracy we would expect to see on the ImageNet shifts, given a (logit) accuracy on the original ImageNet. Given a new model that has accuracy $\text{ACC}(I)$ on ImageNet and average accuracy $\text{ACC}(S)$ on the canonical ImageNet shifts, ER is computed as:

$$\text{ER}(\text{ACC}(S), \text{ACC}(I)) := \text{ACC}(S) - \text{logit}^{-1}\left[\beta_1 \text{logit}\left[\text{ACC}(I)\right] + \beta_0\right]. \quad (1)$$

By fitting a line on the baseline accuracies collected by Taori et al. (2020), we get a slope of $\beta_1 = .76$ and an intercept of $\beta_0 = -1.49$, with a Pearson correlation $r = .99$. This line, along with the baseline models, can be observed in Figure 1.

**Model pool.** We run our analyses across four backbone architectures: ResNet50, ResNet101, ViT-B-16, ViT-B-32 (He et al., 2015; Dosovitskiy et al., 2020). For each architecture, the OpenCLIP repository (Ilharco et al., 2021) contains pretrained CLIP models on various pretraining datasets: the original (unreleased) OpenAI pretraining set (OpenAI, Radford et al. (2021)), YFCC-15M (Thomee et al., 2016; Radford et al., 2021), CC-12M (Changpinyo et al., 2021), LAION-400M, LAION-2B (Schuhmann et al., 2022), and DataComp (Cherti et al., 2023). We load the pretrained vision encoders of all available combinations of architecture and pretraining dataset from OpenCLIP, and construct a zero-shot classification model for ImageNet using the same methodology as Radford et al. (2021). By finetuning each zero-shot model on ImageNet, we obtain classifiers with lower ER than their zero-shot counterparts (Andreassen et al., 2021; Kumar et al., 2022; Wortsman et al., 2022a). Lastly, by training models with an identical architecture on ImageNet from scratch, we obtain models with even lower ER than the finetuned ones (Fang et al., 2022) (details on model finetuning and training can be found in Appendix G). Note that Fang et al. (2022) also obtain CLIP

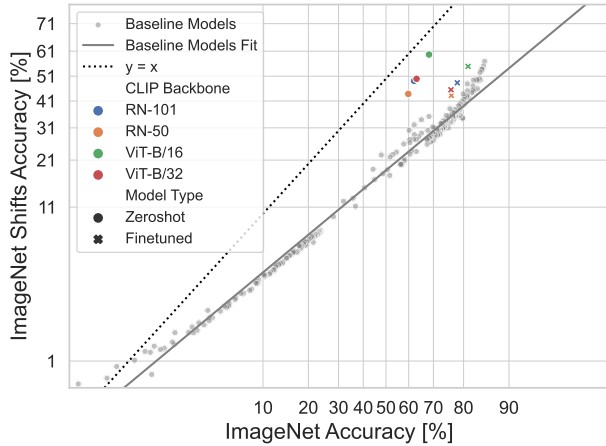

Figure 1: *Accuracies of baseline models and CLIP models on ImageNet and its canonical shifts. The zero-shot CLIP models accuracies are substantially above the baseline fit, i.e., they have high ER. The finetuned CLIP models are closer to the baseline fit, i.e., they have lower ER.*

models with no robustness by training them on ImageNet-Captions, an augmented version of ImageNet where text descriptions of labels have been created from the original labels to allow for CLIP training. However, none of these models achieve ImageNet accuracies beyond 35%, and therefore we choose to restrict our analysis to the models that were trained in a supervised way on ImageNet from scratch when comparing to models with no robustness (rather than CLIP models trained on ImageNet-Captions).

**Results.** We compute the ER with Equation (1) for each model and report the results in Table 1. For each model, we also report the test accuracy they achieve on ImageNet in Appendix A. All of the zero-shot models show robustness, with their ER ranging from 7% for the YFCC models to 37% for the ViT-L-14 . For each model, we observe that finetuning leads to a drop in ER. Finally, the ImageNet supervised models indeed show little to no ER[2].

> **Take-away 1.** In agreement with the ER literature, we find that for each architecture and pre-training set, the zero-shot pretrained CLIP model has the *highest ER*. This ER is significantly decreased after finetuning on ImageNet (*medium ER*). Finally, the same architecture trained from scratch on ImageNet has the *lowest ER* of all three models.

## 3 ROBUST MODELS EXHIBIT OUTLIER FEATURES

In this section, we explain how we identified *outlier features* reflected in *privileged directions* in representation space as a signature of robust models. We first describe our approach, and then explain how outlier features are surfaced and propagate to class logits in the form of privileged directions. Finally, we demonstrate the importance of outlier features and privileged directions on performance by pruning non-privileged directions of the representation space.

**Approach.** We aim to analyze what models with high ER have learned in comparison to models with lower ER. To this end, we compare the features, i.e. the representation space spanned by the activations of the last layer in the encoder, learned by models with different levels of ER. Like Goh et al. (2021), we focus on the last layer since only the output of this layer is used for downstream classification by a linear head computing the ImageNet class logits. Interestingly, a *central kernel alignment* (CKA) analysis in Appendix B reveals that robust models differ from less robust models most consistently in this last layer. We use the ImageNet test set to produce activation vectors $h^{(n)} \in \mathbb{R}^{d_H}$ for each image $x^{(n)} \in \mathbb{R}^{d_X}$ fed to the encoder, where $d_X$, and $d_H \in \mathbb{N}^+$ are respectively the dimension of the input and representation spaces.

---

[2]For the ViT-L-14, we were unable to train a supervised version to convergence from scratch on ImageNet (Dosovitskiy et al. (2020) and He et al. (2022) comment on the difficulties of training such an overparametrized model on ImageNet).

Table 1: *ER for the models in our pool, as calculated by Equation (1). We see that for each backbone and pretraining data, ER decreases from Zero-shot CLIP to Finetuned CLIP and reaches a minimum for ImageNet supervised.*

| Backbone | Pretraining data | Zero-shot CLIP | Finetuned CLIP | ImageNet supervised |
|---|---|---|---|---|
| ResNet50 | OpenAI | 21 % | 8 % | |
| | YFCC-15M | 7 % | 1 % | < 1 % |
| | CC-12M | 14 % | 7 % | |
| ResNet101 | OpenAI | 24 % | 11 % | < 1 % |
| | YFCC-15M | 9 % | 3 % | |
| ViT-B-16 | OpenAI | 31 % | 14 % | |
| | LAION-400M | 24 % | 13 % | 2 % |
| | LAION-2B | 26 % | 12 % | |
| | DataComp | 27 % | 15 % | |
| ViT-B-32 | OpenAI | 24 % | 10 % | |
| | LAION-400M | 24 % | 13 % | 1 % |
| | LAION-2B | 27 % | 12 % | |
| ViT-L-14 | OpenAI | 37 % | 21 % | |
| | LAION-400M | 32 % | 20 % | N.A. |
| | LAION-2B | 32 % | 21 % | |
| | DataComp | 37 % | 24 % | |

**Preliminary observations.** Just by qualitatively observing the distribution of activation vectors, one thing that immediately stands out is the fact that some components $i \in [d_H]$ are much larger than the average activation: $h_i^{(n)} \gg d_H^{-1} \sum_{j=1}^{d_H} h_j^{(n)}$. A similar phenomenon has recently been observed by Dettmers et al. (2022) in large language models (LLMs). Such features, whose activation is substantially more important than average, were coined as *outlier features*. Subsequent work by Elhage et al. (2023) introduced a simple way to surface these outlier features, through a metric called *activation kurtosis*. We now use this criterion to quantitatively analyze the features of robust models.

**Activation kurtosis.** Following Elhage et al. (2023), we measure the *activation kurtosis* to quantitatively evaluate the presence of outlier features in a model. The activation kurtosis is computed over all the components of an activation vector $h^{(n)}$, and averaged over $N$ activation vectors:

$$\overline{\text{kurtosis}} := \frac{1}{N} \sum_{n=1}^{N} \frac{1}{d_H} \sum_{i=1}^{d_H} \left[ \frac{h_i^{(n)} - \mu\left(h^{(n)}\right)}{\sigma\left(h^{(n)}\right)} \right]^4, \tag{2}$$

where $\mu(h) := d_H^{-1} \sum_{i=1}^{d_H} h_i$ and $\sigma^2(h) := d_H^{-1} \sum_{i=1}^{d_H} [h_i - \mu(h)]^2$. As explained by Elhage et al. (2023), $\overline{\text{kurtosis}} \gg 3$ indicates the presence of outlier features in the studied direction ($\overline{\text{kurtosis}} = 3$ being the kurtosis of an isotropic Gaussian).

We report the average kurtosis over the ImageNet test set in Table 2 for each architecture and across the various levels of ER. We observe two things. Firstly, all models with high robustness have outlier features, as indicated by their $\overline{\text{kurtosis}} \gg 3$. Secondly, the kurtosis, like the ER, drops when finetuning on ImageNet. The values $\overline{\text{kurtosis}} \approx 3$ obtained for finetuned and supervised models suggests the absence of outlier features in the less robust models.

**Privileged directions in representation space.** The strong activation of outlier features in robust models does not necessarily explain the performances of these models. Indeed, it is perfectly possible that outlier features are ignored by the linear head computing the class logits based on the activation vectors, e.g. if they are part of $\ker W$, the null space of the weight matrix $W$ of the linear classification head.

Thus, to assess whether outlier features are of importance, we now introduce the notion of *privileged directions* of the representation space $\mathbb{R}^{d_H}$, as an instance of a generalized form of outlier features. While outlier features are studied in the canonical basis $\{e_1, \ldots, e_{d_H}\}$ (we can write $h_i = \text{Proj}_{e_i}(h)$), they can be generalized to be any set of directions of the representation space that receive a projection substantially above average (for an illustration, see Appendix E).

Table 2: *Results of the kurtosis analysis showing outlier features present in robust models, but disappearing as soon as models are finetuned or trained on ImageNet. Values calculated according to Equation (2) over all ImageNet test examples.*

| Backbone | Pretraining data | Zero-shot CLIP *Highest ER* | Finetuned CLIP *Medium ER* | ImageNet supervised *Lowest ER* |
|---|---|---|---|---|
| ResNet50 | OpenAI | 73.6 | 4.6 | |
| | YFCC-15M | 8.0 | 3.9 | 2.9 |
| | CC-12M | 8.2 | 4.0 | |
| ResNet101 | OpenAI | 32.1 | 3.4 | |
| | YFCC-15M | 7.3 | 3.5 | 2.9 |
| ViT-B-16 | OpenAI | 81.0 | 3.4 | |
| | LAION-400M | 10.9 | 3.4 | |
| | LAION-2B | 25.2 | 3.7 | 4.2 |
| | DataComp | 24.3 | 3.1 | |
| ViT-B-32 | OpenAI | 74.6 | 3.1 | |
| | LAION-400M | 19.5 | 3.1 | 3.9 |
| | LAION-2B | 12.1 | 3.2 | |
| ViT-L-14 | OpenAI | 60.8 | 4.6 | |
| | LAION-400M | 20.3 | 5.2 | |
| | LAION-2B | 66.2 | 6.9 | N.A. |
| | DataComp | 37.4 | 4.6 | |

We choose to focus on the directions that are important for the computation of logits by the linear head $W$, namely *right singular vectors* (RSV) of $W$. These can be identified by performing a singular value decomposition (SVD) of $W$, which can be written as:

$$W = \sum_{i=1}^{\text{rank}(W)} \sigma_i \cdot u_i v_i^{\mathsf{T}}, \tag{3}$$

where $\sigma_i \in \mathbb{R}^+$, $u_i \in \mathbb{R}^{d_Y}$ and $v_i \in \mathbb{R}^{d_H}$ respectively correspond to *singular values* (SV), *left singular vectors* (LSV) and RSVs of $W$. In this decomposition, each RSV $v_i$ corresponds to a direction in representation space that is mapped to the logits encoded in the LSV $v_i$. Since both of these vectors are normalized $\|u_i\| = \|v_i\| = 1$, the importance of the direction $v_i$ for $W$ is reflected by the SV $\sigma_i$. Note that the SV $\sigma_i$ by itself *does not* refer to the model's encoder activations. How can we measure if the direction $v_i$ is typically given an important activation by the encoder? To measure that, we propose to measure the average cosine similarity of activation vectors $h^{(n)}$ with this direction. We aggregate these two sources of importance in a unified metric:

$$\text{Importance}(i) = \underbrace{\frac{\sigma_i}{\sum_{j=1}^{\text{rank}(W)} \sigma_j}}_{\text{Classification head importance}} \cdot \underbrace{\sum_{n=1}^{N} \frac{|\cos(v_i, h^{(n)})|}{N}}_{\text{Encoder importance}}. \tag{4}$$

Note that this metric is defined so that $0 < \text{Importance}(i) < 1$. With this metric, we can measure to what extent the presence of outlier features induces privileged directions in representation space. If such privileged directions exist, we expect some singular directions $v_i$ to have an $\text{Importance}(i)$ substantially higher than average, i.e. with $\text{Importance}(i) \gg d_H^{-1} \sum_{j=1}^{d_H} \text{Importance}(j)$. We thus can identify privileged directions as the RSVs associated to outlier values in the importance scores.

Indeed, in Figure 2 where we plot the distribution of importance scores over all RSVs, we can observe the existence of such privilege directions. We notice that all the robust zero-shot models have at least one privileged direction. For the less robust finetuned models, these privileged directions still exist, but with lower importance score. This indicates that finetuning de-emphasizes privileged directions. Finally, the importance distributions of non-robust supervised models have no privileged directions. Interestingly, our privileged directions analysis allows us to distinguish between the differences in ER of finetuned and supervised models, which is not the case for activation kurtosis.

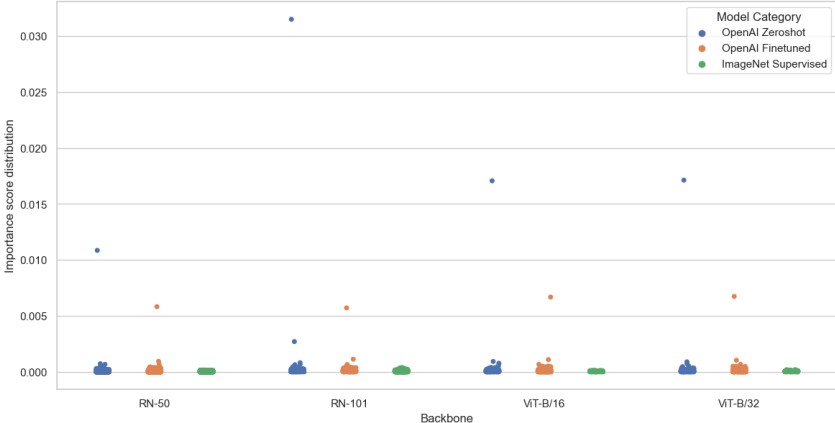

Figure 2: *Distribution of importance scores as calculated by Equation (4) over all RSV in the representation space. Zero-shot models have longer tails, associated to privileged directions. Finetuned models still exhibit privileged directions, but with lower importance. Supervised models have no such privileged directions. Analogous results for remaining models can be found in Appendix D.1*

> **Take-away 2.** Robust models exhibit outlier features that induce privileged directions in their representation spaces. This appears to be a signature of models with high ER.

Note that previous work on LLMs found that outlier features also have positive effects on model pruning: Sun et al. (2023) found that LLMs with outlier features can be efficiently pruned by retaining features with larger activations. We also found some evidence of this kind when pruning latent directions (see Appendix F).

## 4 ROBUST MODELS ENCODE MORE CONCEPTS

In this section, we explain how we found that models with high ER encode more concepts. We first describe our approach, and then discuss the concepts encoded in the privileged directions identified in the previous section. We then show that robust models encode more unique concepts. Lastly, we explain how this leads to polysemanticity.

**Approach.** With the discovery of privileged directions in the representation spaces of models with high ER, it is legitimate to ask what type of information these directions encode. More generally, are there differences in the way robust models encode human concepts? To answer these questions, we use *concept probing*. This approach was introduced by Bau et al. (2017), along with the *Broden dataset*. This dataset consists of $63,305$ images illustrating $C = 1,197$ concepts, including scenes (e.g. street), objects (e.g. flower), parts (e.g. headboard), textures (e.g. swirly), colors (e.g. pink) and materials (e.g. metal). Note that several concepts can be present in each image. For each concept $c \in [C]$, we construct a set of positive images $\mathcal{P}^c$ (images that contain the concept) and negative images $\mathcal{N}^c$ (images that do not contain the concept). In the following, we shall consider balanced concept sets: $|\mathcal{P}^c| = |\mathcal{N}^c|$. Concept probing consists in determining if activations in a given direction of the representation space discriminate between $\mathcal{P}^c$ and $\mathcal{N}^c$.

**Assigning concepts to directions.** We are interested in assigning concepts to each RSV $v_i$ of the linear head matrix $W$. To determine whether a representation space direction enables the identification of a concept $c \in [C]$, we proceed as follows. For each activation vector $h^{(c,n)}$ associated to positive images $x^{(c,n)} \in \mathcal{P}^c$, we compute the projection $\mathrm{Proj}_{v_i}(h^{(c,n)})$ on the RSV. We perform the same computations for the projections $\mathrm{Proj}_{v_i}(h^{(\neg c,n)})$ of negative images $x^{(\neg c,n)} \in \mathcal{N}^c$. If the direction $v_i$ discriminates between concept negatives and positives, we expect a separation between these projections: $\mathrm{Proj}_{v_i}(h^{(c,n)}) \neq \mathrm{Proj}_{v_i}(h^{(\neg c,n)})$. In other words, we expect the projections on $v_i$ to be a good classifier to predict the presence of $c$. Following Cuadros et al. (2022), we measure

the average precision $\mathtt{AP}_i^c$ of this classifier to determine whether the concept is encoded in direction $v_i$. We set a threshold $\mathtt{AP}_i^c \geq 0.9$ to establish that the concept $c$ is encoded in $v_i$[3].

**Interpreting privileged directions.** We look at the concepts with highest AP in the most privileged direction of each model represented in Figure 2. All the concepts discussed below are encoded with AP $> 0.9$. Interestingly, the most privileged direction of both zero-shot OpenAI ViTs encode the same top-3 concepts: *meshed, flecked* and *perforated*. The most privileged direction of the ResNet50 also encodes concepts related to textures, with the *knitted* and *chequered* concepts. The most privileged direction of the ResNet101 encodes concepts with high colour contrasts, with the *moon bounce*, *inflatable bounce game* and *ball pit* concepts. We note that all these concepts describe regular alternating patterns, either through the presence/absence of holes or through the variation of colours. An interesting parallel can be drawn with the work of Bondarenko et al. (2023), which found that outlier features in language models assign most of their mass to separator tokens (such as the end of sentence token).

Let us now discuss the concepts encoded in the privileged directions of finetuned models. It is striking that finetuning replaces the above texture-related concepts by more concrete concepts. After finetuning, the concept that are best encoded in the privileged directions are *martial art gym* for the ViT-B/16, *tennis court* for the ViT-B/32, *mountain pass* for the ResNet50 and *flight of stairs* for the ResNet101. All of these concepts are substantially less generic than the ones encoded in the zero-shot models. We do not discuss supervised models here, as Figure 2 demonstrated that they have no privileged direction. For completeness, we also report the top-3 concepts for the models not shown in Figure 2 in Appendix D.3. We note that the above discussion generalizes well to the other pretraining sets.

> **Take-away 3.** Privileged directions of zero-shot models encode generic texture information. Finetuning replaces these generic concepts in privileged directions by more concrete concepts.

**Number of unique concepts.** Let us now discuss the representation spaces of various models beyond privileged directions. A first way to characterize a representation space as a whole is to simply count the number of unique concepts they encode. In other words, for the representation space of each model, we evaluate

$$N_{\text{unique}} := |\{c \in [C] \mid \mathtt{AP}_i^c \geq 0.9 \text{ for some } i \in [d_H]\}|. \tag{5}$$

We report the number of unique concepts encoded in each type of model from our pool in Table 3. We notice that zero-shot models encode substantially more concepts than their finetuned and supervised counterparts. To further compare the set of concepts encoded in each model, we produce their *Venn diagrams* in Figure 3. In each case, the most significant section of the Venn diagrams is the overlap between all 3 model types (this ranges from 237 concepts for RN-101 to 516 concepts for ViT-B/16). This suggests that all 3 models share a large pool of features that are useful for each respective task the models were trained on. Beyond this strong overlap, we note that the concepts of finetuned models are mostly subsets of the concepts encoded in zero-shot models. This is somewhat expected, as finetuned models greatly benefit from features of pretrained models, which explains their medium ER. In agreement with Table 3, we observe that zero-shot models encode many concepts that are unknown to finetuned and supervised models (this ranges from 77 concepts for ViT-B/16 to 105 concepts for RN-50). This large addition of concepts make the representation spaces of zero-shot models richer.

**Connection to polysemanticity.** A large number of encoded concepts can come at the cost of interpretability. As explained by Olah et al. (2020), superposing many concepts in a given representation space creates *polysemantic features*. Those features correspond to directions of the representation spaces that encode several unrelated concepts, which makes the interpretation of such features challenging. Polysemantic features are typically identified by using feature visualization to construct images that maximally activate the unit (neuron / representation space direction) of interest (Olah et al., 2017). A manual inspection of these images permits to identify that several concepts are present in the image maximizing the activation of the unit of interest.

Clearly, a manual inspection of feature visualizations for each RSV $v_i$ of each model in our pool would be prohibitively expensive. For this reason, we use a *proxy* for polysemanticity based on

---

[3]All the below conclusions still hold if we change this threshold to e.g. .85.

Table 3: *Results of the unique concept analysis, showing total # of unique Broden concepts encoded in last layers, as in Equation* (5). *Zero-shot models encode substantially more concepts.*

| Backbone | Pretraining data | Zero-shot CLIP
*Highest ER* | Finetuned CLIP
*Medium ER* | ImageNet supervised
*Lowest ER* |
|---|---|---|---|---|
| ResNet50 | OpenAI | 507 | 311 | |
| | YFCC-15M | 652 | 602 | 418 |
| | CC-12M | 647 | 584 | |
| ResNet101 | OpenAI | 489 | 323 | 397 |
| | YFCC-15M | 604 | 568 | |
| ViT-B-16 | OpenAI | 702 | 555 | |
| | LAION-400M | 672 | 574 | 635 |
| | LAION-2B | 733 | 582 | |
| | DataComp | 701 | 527 | |
| ViT-B-32 | OpenAI | 689 | 557 | |
| | LAION-400M | 676 | 550 | 559 |
| | LAION-2B | 672 | 527 | |
| ViT-L-14 | OpenAI | 704 | 623 | |
| | LAION-400M | 683 | 613 | N.A. |
| | LAION-2B | 704 | 633 | |
| | DataComp | 684 | 619 | |

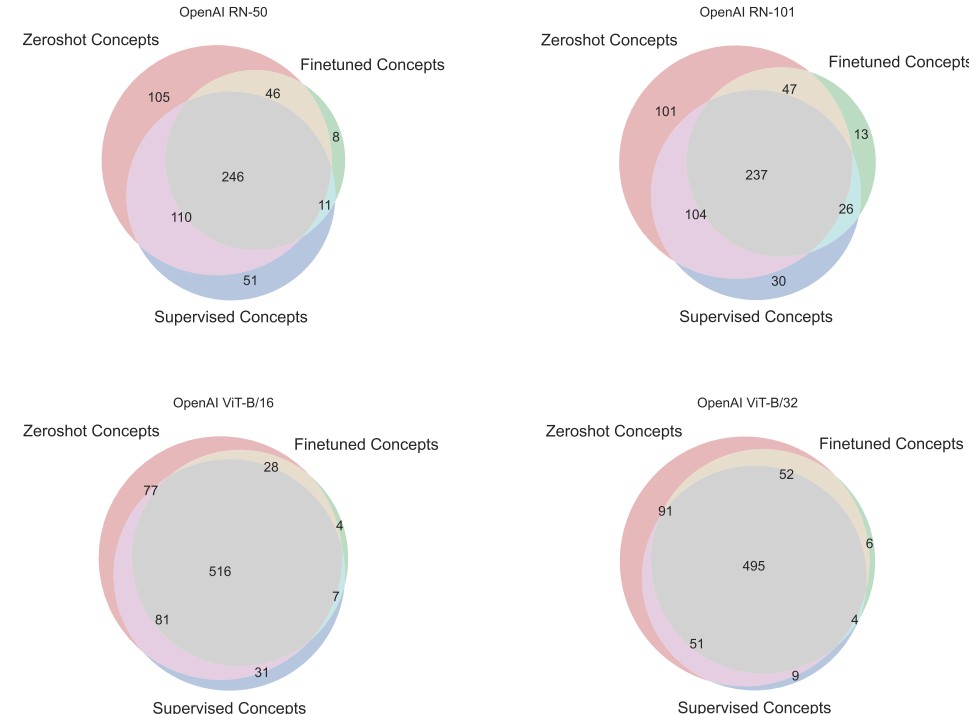

Figure 3: *Overlap between the concepts encoded in the representation space of different models for each OpenAI models. Zero-shot models encode many concepts not encoded other models.*

the Broden dataset. For each RSV $v_i$, we count the number of concepts encoded in the corresponding direction of the representation space $N_{\text{concept}}(i) := |\{c \in [C] \mid \text{AP}_i^c \geq 0.9\}|$. The higher this number is, the more likely it is that the feature corresponding to $v_i$ is polysemantic. As a measure of polysemanticity for the model, we simply average this number over all singular vectors: polysemanticity $:= d_H^{-1} \sum_{i=1}^{d_H} N_{\text{concept}}(i)$. By measuring this number of all zero-shot models, we found that this ranges from polysemanticity $= 3$ for the OpenAI ResNet 50 to polysemanticity $= 16$ for the LAION-2B ViT-B/16. By looking at the complete results in Appendix D.4, we also note that most zero-shot models are on the higher side of this range, with

typically more than 10 concepts encoded in one direction on average. Since the Broden dataset has no duplicate concepts, we deduce that these models are highly polysemantic.

> **Take-away 4.** The increased amount of concepts encoded in zero-shot models is another signature of ER. The large number of concepts encoded in zero-shot models makes these models polysemantic.

## 5 RELATED WORK

**Interpretability and CLIP models.** A number of works previously studied CLIP from a model-centric/interpretability perspective. We can broadly divide these works into 2 categories. (1) The first body of work, like ours, uses interpretability methods to gain a better understanding of CLIP. For instance, Li et al. (2022) analyze saliency maps of CLIP and found that the model tends to focus on the background in images. Goh et al. (2021) analyzed CLIP ResNets and found *multimodal neurons*, that respond to the presence of a concept in many different settings. (2) The second body of work leverages CLIP to explain other models. For instance, Jain et al. (2022) use CLIP to label hard examples that are localized as a direction in any model's representation space. Similarly, Oikarinen & Weng (2022) use CLIP to label neurons by aligning their activation patterns with concept activation patterns on a probing set of examples. To the best of our knowledge, our work is the first to leverage interpretability to better understand robustness of CLIP to natural distribution shifts.

**Outlier features in foundation models.** Outlier features in foundation models were first discovered in LLMs by Dettmers et al. (2022). Those features are found to have an adverse effect on the model quantization. The reason for which outlier features appear in LLMs is yet unknown. Elhage et al. (2023) investigated several possibles causes (such as layer normalization), but found no conclusive explanation. They conclude that the emergence of outlier features is most likely a relic of Adam optimization. Bondarenko et al. (2023) found that outlier features in transformers assign most of their mass to separator tokens and that modifying the attention mechanism (by clipping the softmax and using gated attention) decreases the amount of outlier features learned during pretraining. To the best of our knowledge, our work is the first to discuss outlier features outside of language models.

## 6 DISCUSSION

With a thorough investigation of the representation spaces of robust CLIP models, we found two signatures that set them apart from other models. (1) Robust CLIP models have outlier features, which induce privileged directions in these model's representation spaces. (2) Robust CLIP models encode substantially more concepts in their representation space. While this makes the representation spaces of these models richer, this also induces a high polysemanticity in their features, making their interpretation challenging. Crucially, these observations distinguish a large range of robust models with high ER from models with lower ER. In fact, an additional analysis in Appendix I shows that the kurtosis and the number of unique encoded concepts closely tracks the ER metric when interpolating between zero-shot and finetuned CLIP models in weight space. Therefore, we posit that these two signatures offer good proxies for ER, with the advantage of being easy to compute and not requiring access the shifted distributions. Interestingly, we can also validate these signatures on non-CLIP models such as the multimodal CoCa (Yu et al., 2022) models or the pure vision NoCLIP (Fang et al., 2022) models in Appendix C.

Beyond progressing the understanding of how ER manifests in CLIP models, we believe our work opens up many interesting research directions. A first one would be to extend the analysis of this work to dataset shifts beyond the ImageNet family, to see if the signatures are relevant beyond the much investigated ImageNet shifts. Another interesting research direction would be to further investigate a potential trade-off between robustness and interpretability. While our results suggest that all the models with high robustness tend to be polysemantic, it is not clear if these two characteristics can be disentangled. Finally, it would be interesting to extend the scope of investigation to the pretraining phase of these models, possibly by tracking our signatures.

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

## A   IMAGENET ACCURACIES FOR OUR MODEL POOL

In Table 4 we show the top-1 accuracies the models analysed in the main part of the paper achieve on the ImageNet test set.

Table 4: *ImageNet test set accuracies for the models under investigation*

| Backbone | Pretraining data | Zero-shot CLIP | Finetuned CLIP | ImageNet supervised |
|---|---|---|---|---|
| ResNet50 | OpenAI | 60 % | 76 % | |
| | YFCC-15M | 32 % | 69 % | 70 % |
| | CC-12M | 36 % | 69 % | |
| ResNet101 | OpenAI | 62 % | 78 % | 71 % |
| | YFCC-15M | 34 % | 72 % | |
| ViT-B-16 | OpenAI | 68 % | 81 % | |
| | LAION-400M | 67 % | 80 % | 80 % |
| | LAION-2B | 70 % | 81 % | |
| | DataComp | 63 % | 78 % | |
| ViT-B-32 | OpenAI | 63 % | 78 % | |
| | LAION-400M | 60 % | 76 % | 75 % |
| | LAION-2B | 66 % | 76 % | |
| ViT-L-14 | OpenAI | 75 % | 85 % | |
| | LAION-400M | 73 % | 84 % | N.A. |
| | LAION-2B | 74 % | 84 % | |
| | DataComp | 79 % | 85 % | |

## B   ZERO-SHOT AND FINETUNED MODELS' DIFFERENCES ARE LOCALIZED

In this appendix, we apply central kernel alignment (CKA) to identify where changes between more robust and less robust models occur. Kornblith et al. (2019) introduce the CKA metric to quantify the degree of similarity between the activation patterns of two neural network layers. It takes two batch of activation vectors $a$ and $b$, it computes their normalized similarity in terms of the Hilbert-Schmidt Independence Criterion (HSIC, Gretton et al. (2005)):

$$\text{CKA}(\boldsymbol{a}, \boldsymbol{b}) := \frac{\text{HSIC}(\boldsymbol{a}, \boldsymbol{b})}{\sqrt{\text{HSIC}(\boldsymbol{a}, \boldsymbol{a})}\sqrt{\text{HSIC}(\boldsymbol{b}, \boldsymbol{b})}}$$

We use the PyTorch-Model-Compare package (Subramanian, 2021) to compute this metric between the activation vectors of zero-shot models and their finetuned counterparts for each layer in the backbone. The results are shown in Figure 4 and Figure 5. Across architectures and pretraining sets, we find that there is often a large drop in CKA between zero-shot and finetuned models occurring in the last layer. This makes the activations in the last layer a particularly interesting layer to analyse when investigating ER, as finetuned models typically have only half the ER of their zero-shot counterpart (see Table 1).

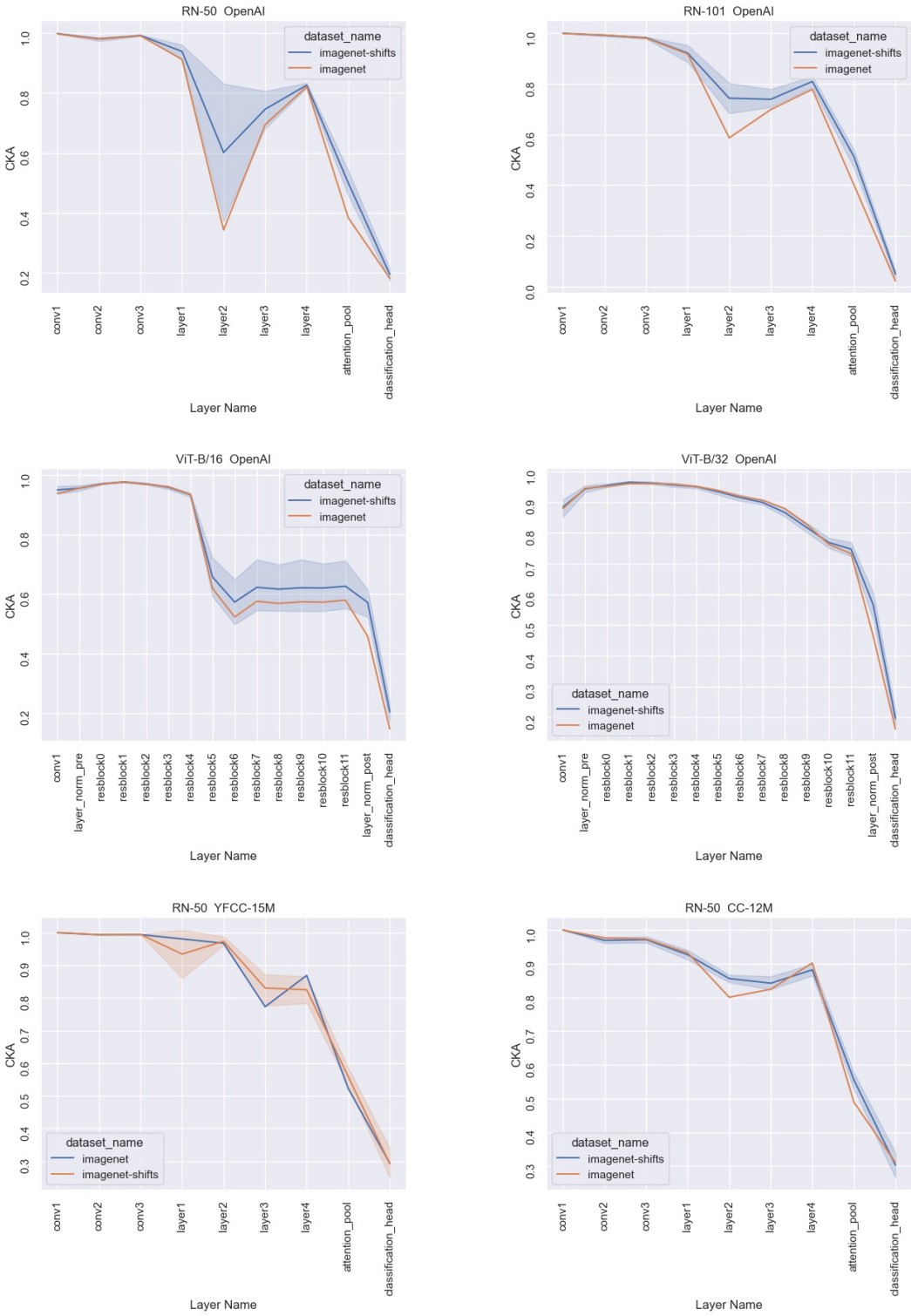

Figure 4: *Result of layer by layer CKA comparison between zero-shot CLIP and its counterpart that was finetuned on ImageNet for various backbones and pretraining sets (Part 1). In orange, CKA between activation vectors on ImageNet test set. In blue, CKA between activation vectors on shifted ImageNet sets (average as solid line, standard deviation in shaded blue). Typically, we see large drops of CKA in the last layer.*

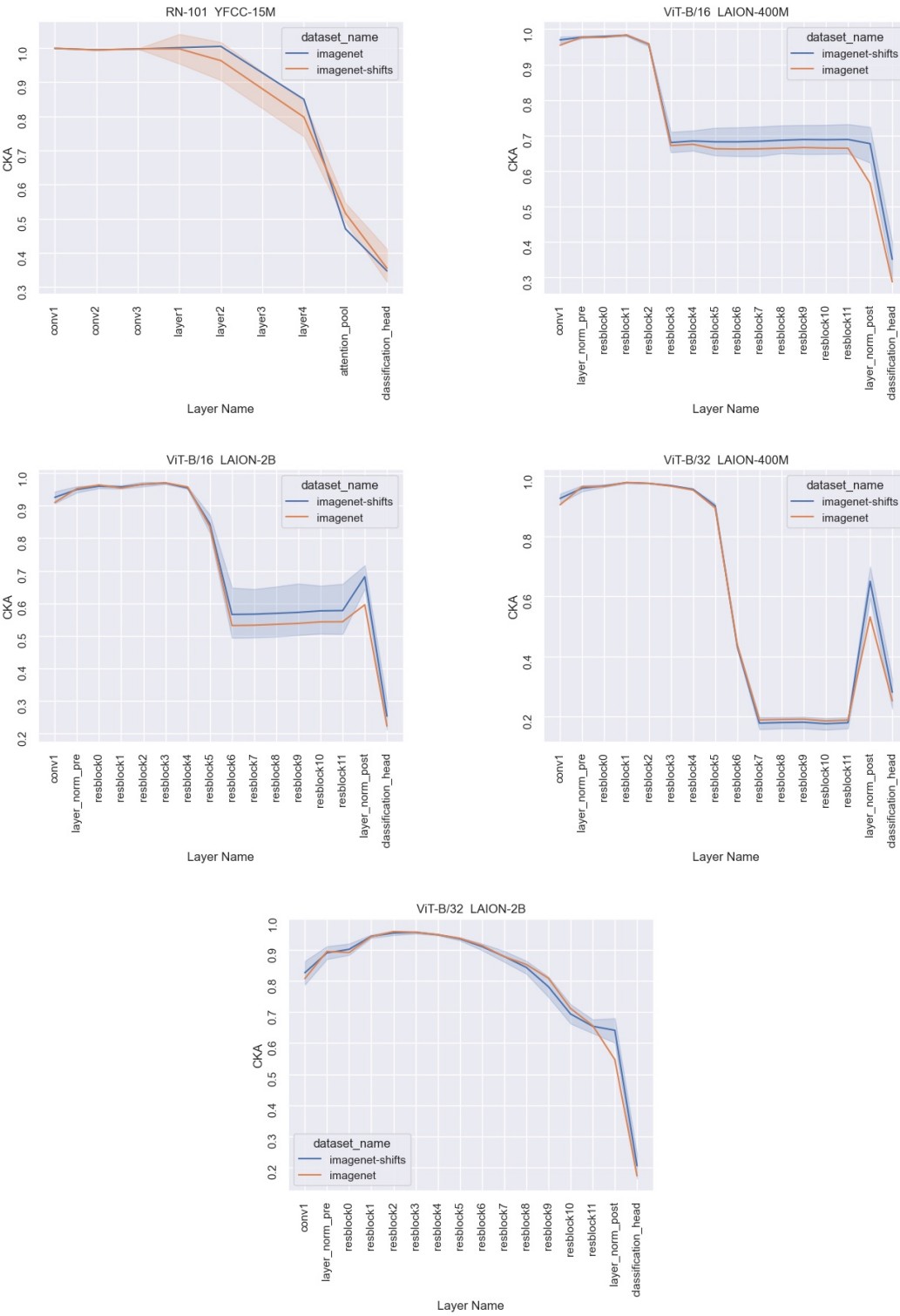

Figure 5: *Result of layer by layer CKA comparison between zero-shot CLIP and its counterpart that was finetuned on ImageNet for various backbones and pretraining sets (Part 2). In orange, CKA between activation vectors on ImageNet test set. In blue, CKA between activation vectors on shifted ImageNet sets (average as solid line, standard deviation in shaded blue). Typically, we see large drops of CKA in the last layer.*

# C ROBUSTNESS SIGNATURES IN NON-CLIP MODELS

## C.1 COCA MODELS

In addition to the CLIP models investigated in the main paper, we below investigate CoCa models (Yu et al., 2022) as another set of robust multimodal models which do not fall into the CLIP family. We see that our findings extend to these non-CLIP multimodal models as well.

First, in Table 5, we confirm that these models have high effective robustness when used as zero-shot classifiers. As for the other models in the main part of the paper, we observe that finetuning on ImageNet decreases the effective robustness of these classifiers.

Table 5: *ER for the models of the CoCa family, as calculated by Equation (1) (accuracies on ImageNet shown in brackets). We see that also for these models, ER decreases from Zero-shot to Finetuned.*

| Backbone | Pretraining data | CoCa Zero-shot | CoCa Finetuned |
|----------|------------------|----------------|----------------|
| ViT-B-32 | LAION-2B | 24% (64%) | 14% (76%) |
| ViT-L-14 | LAION-2B | 34% (76%) | 21% (84%) |

Next, in Table 6, we show that all the zero-shot models have high kurtosis, which implies the existence of outlier features in their representation space. Additionally, we show that finetuning again decreases the kurtosis.

Table 6: *Results of the kurtosis analysis showing outlier features present also in robust models with ViT-L-14 backbone or of CoCa family, but disappearing as soon as models are finetuned. Values calculated according to Equation (2) over all ImageNet test examples.*

| Backbone | Pretraining data | CoCa Zero-shot | CoCa Finetuned |
|----------|------------------|----------------|----------------|
| ViT-B-32 | LAION-2B | 12.0 | 3.6 |
| ViT-L-14 | LAION-2B | 15.5 | 4.6 |

Finally, in Table 7, we check that zero-shot model encodes more concepts. Again, we see that finetuning removes some concepts from the model's representation space.

Table 7: *Results of the unique concept analysis, showing total # of unique Broden concepts encoded in last layers, as in Equation (5). Zero-shot models encode substantially more concepts.*

| Backbone | Pretraining data | CoCa Zero-shot | CoCa Finetuned |
|----------|------------------|----------------|----------------|
| ViT-B-32 | LAION-2B | 674 | 530 |
| ViT-L-14 | LAION-2B | 747 | 629 |

## C.2 NOCLIP MODELS

Interestingly, our findings on outlier features as a signature of ER do not necessarily only hold for multimodal models, but can also be found in pure vision models, as the example of NoCLIP illustrates. NoCLIP was introduced by Fang et al. (2022) as a model that exhibits positive ER without the multimodal pretraining of CLIP models in their quest to find the reason for CLIPs remarkable ER (deducing from NoCLIP that the reason does not lie in the multimodality / language supervision). It was obtained by pretraining a VIT-B-16 model in a SimCLR (Chen et al., 2020) fashion on the YFCC-15M dataset, and then finetuning it in a supervised way on a subset of YFCC-15M that can be assigned ImageNet class labels. We load it from OpenCLIP (Ilharco et al., 2021).

As can be seen in Table 8, it achieves 4% ER, which is similar to the ER that a VIT-B-16 CLIP model achieves when pretrained on YFCC-15M (Fang et al., 2022). While it does not exhibit excess kurtosis, it does have a fairly high number of unique concepts encoded in its features. Importantly,

the importance analysis of RSVs in Figure 6 surfaces that it indeed has privileged directions, like other robust do.

Thus, our finding that outlier features in the form of strong privileged directions in feature representations space are a signature of robust multimodal models also holds for a pure vision model.

Table 8: *Effective Robustness and our metrics for NoCLIP.*

| Model | ImageNet ACC | ER | Activation kurtosis | # unique concepts |
|---|---|---|---|---|
| NoCLIP SimCLR | 35 % | 4 % | 3.4 | 621 |

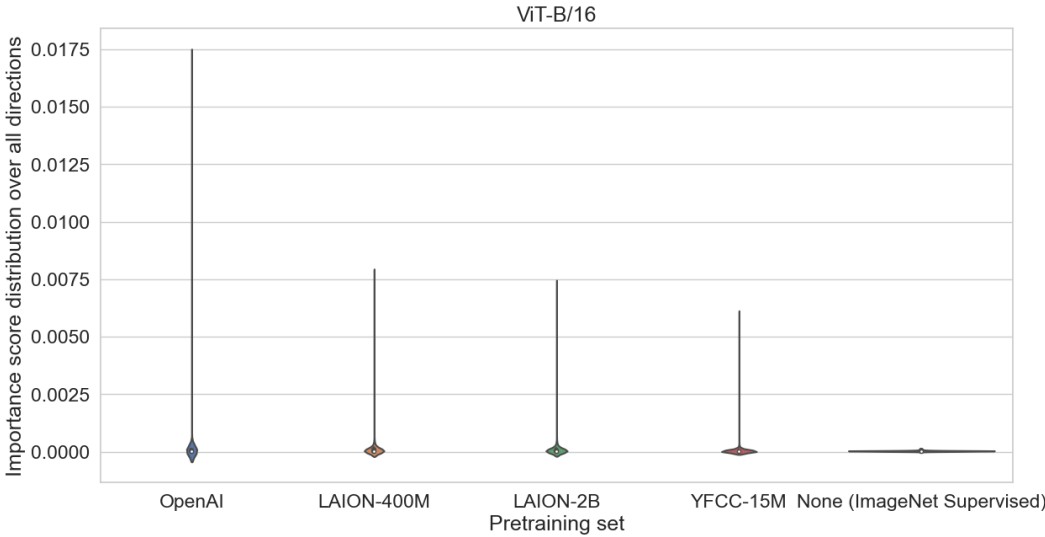

Figure 6: *Comparison of privileged importance distribution between NoCLIP and other robust zero-shot models. NoCLIP corresponds to the model pretrained on YFCC-15M in this plot.*

# D    FURTHER EXPERIMENT RESULTS

## D.1    PRIVILEGED DIRECTION ANALYSIS FOR EACH MODEL

In Figure 7 we show the analysis of privileged directions in representation space from Figure 2 in the main paper repeated for zero-shot and finetuned models for the OpenAI pretraining set and the remaining pretraining sets for each of the four backbones. We don't show the results for ImageNet Supervised as they are already shown in Figure 2 for each of the backbones.

Qualitatively, our finding from the main paper is confirmed across the remaining models we investigate: zero-shot models have one strongly privileged direction and finetuned models still exhibit a privileged direction, but with lower importance. The only exception to this pattern are the YFCC-15M pretrained ResNets, for which the privileged direction is stronger for the finetuned model. We suspect that this has to do with the fact that the zero-shot model starts from a very low accuracy on ImageNet test to begin with (32% and 34%, see Table 4).

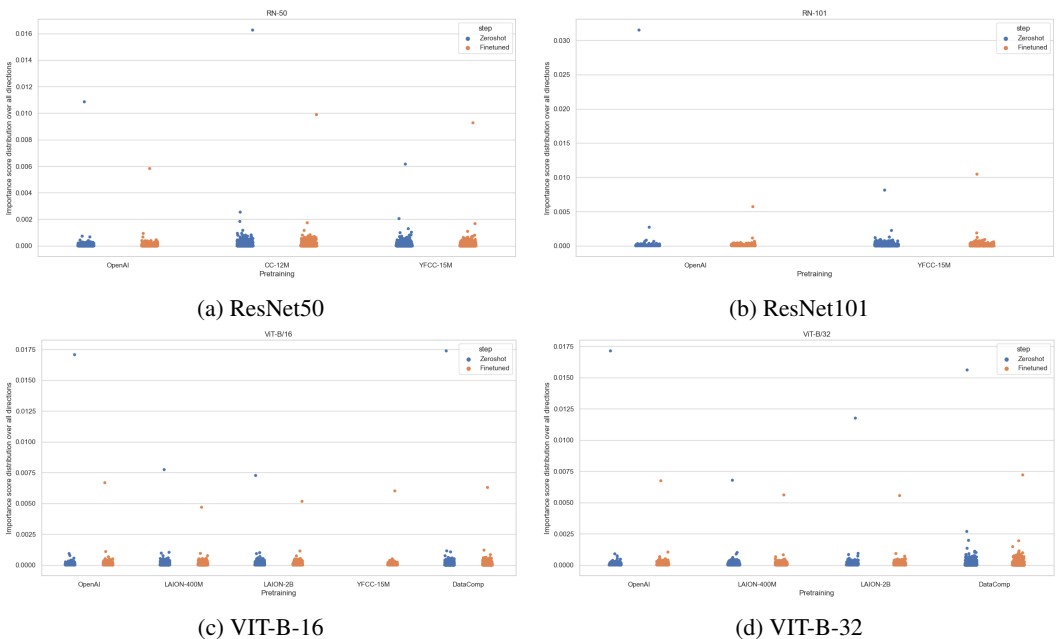

(a) ResNet50

(b) ResNet101

(c) VIT-B-16

(d) VIT-B-32

Figure 7: Distribution of importance over all RSV in the representation space for zero-shot and finetuned models. Zero-shot models have one strongly privileged direction and sometimes a few more mildly privileged directions. Finetuned models still exhibit one strong privileged direction, but with lower importance than the zero-shot models, and sometimes a few mildly privileged directions.

## D.2 PRUNING ANALYSIS FOR EACH MODEL

Figure 8 and Figure 9 show the effect of gradually pruning the least important SV of $W$ on ER and ACC for all models that were not pretrained on the OpenAI pretraining dataset, similar to the analysis shown in Figure 10 in the main paper for the OpenAI pretrained models.

Qualitatively, our finding from the main paper is confirmed across the remaining models we investigate: A small subset (typically around 20%) of privileged directions in representation space explain the high ER of zero-shot models. The remaining directions can be pruned without significantly impacting neither ER nor ACC.

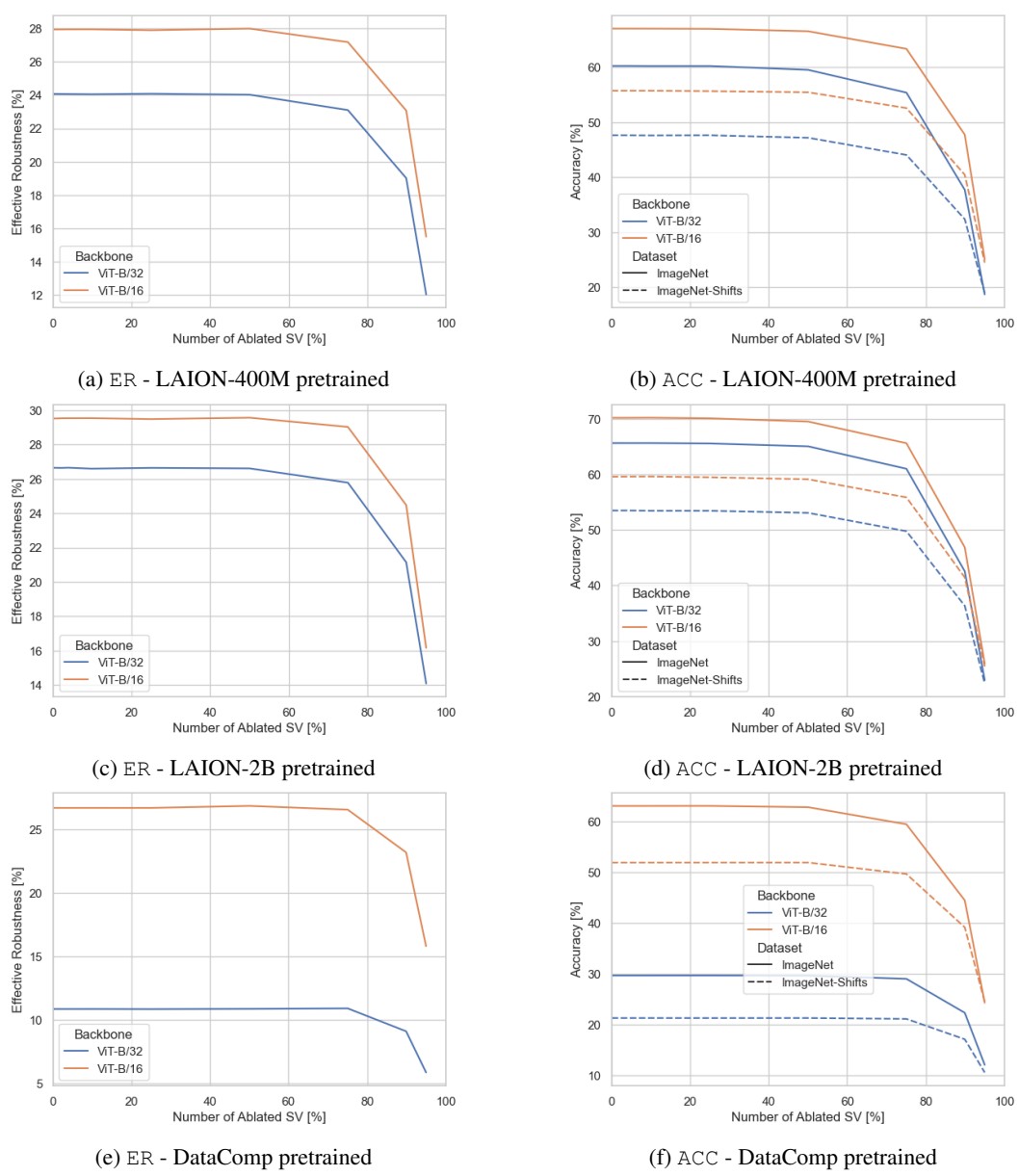

(a) ER - LAION-400M pretrained

(b) ACC - LAION-400M pretrained

(c) ER - LAION-2B pretrained

(d) ACC - LAION-2B pretrained

(e) ER - DataComp pretrained

(f) ACC - DataComp pretrained

Figure 8: *Effect of gradually pruning the least important SV of W on* ER *and* ACC *for LAION-400M, LAION-2B, and DataComp pretrained models.*

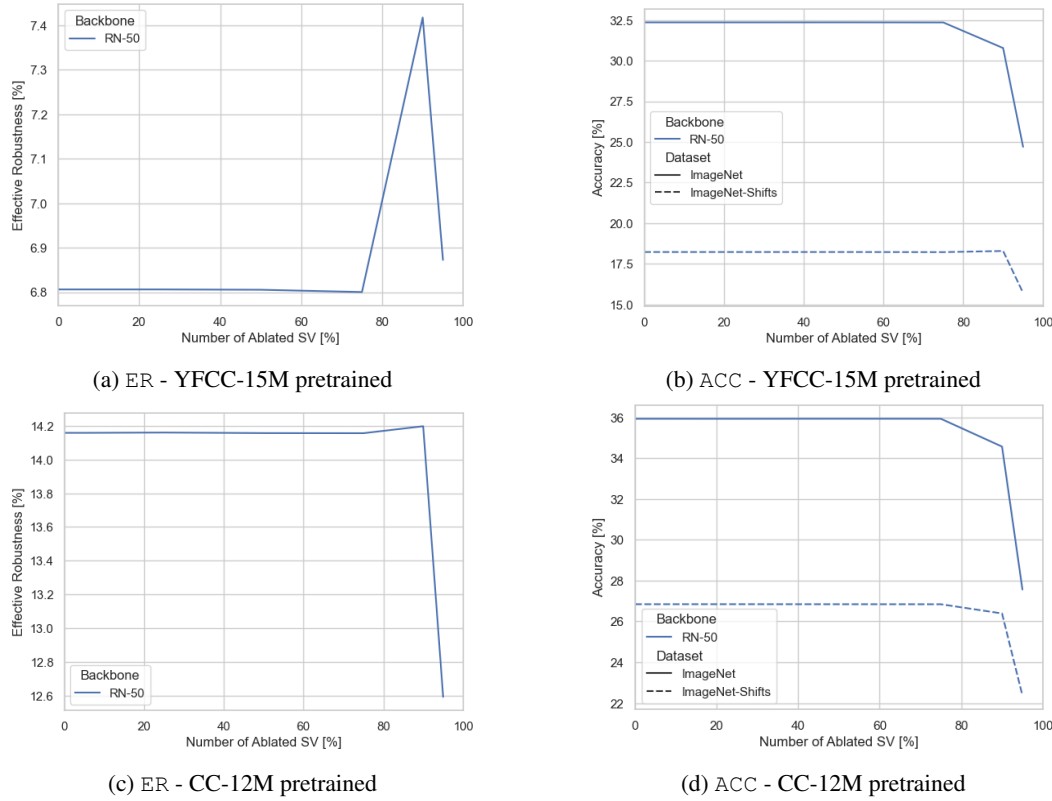

(a) ER - YFCC-15M pretrained

(b) ACC - YFCC-15M pretrained

(c) ER - CC-12M pretrained

(d) ACC - CC-12M pretrained

Figure 9: *Effect of gradually pruning the least important SV of W on ER and ACC for YFCC-15M and CC-12M pretrained models.*

### D.3 TOP-3 CONCEPTS IN MOST DOMINANT DIRECTION FOR EACH MODEL

We look at the concepts with highest AP in the most privileged direction of each model represented in Figure 7, similar to what we did in Section 4 for the most privileged direction of each model in Figure 2. The results are shown in Table 9. Again, we observe that for the majority of cases, the robust zero-shot models encode concepts related to textures as their top concepts encoded along the privileged directions (e.g., *scaly*, *meshed*, or *matted*), while the less robust finetuned models encode more concrete concepts (e.g., *carrousel*, *book stand*, or *pantry*).

Table 9: *Top-3 concepts with highest AP encoded in the most privileged direction of each model. For each concept, the AP is included in brackets.*

| Backbone | Step Pretraining | Top-1 Concept Finetuned | Zeroshot | Top-2 Concept Finetuned | Zeroshot | Top-3 Concept Finetuned | Zeroshot |
|---|---|---|---|---|---|---|---|
| RN-101 | OpenAI | flight of stairs natural s ( 0.9) | moon bounce s ( 0.99) | movie theater indoor s ( 0.88) | inflatable bounce game ( 0.99) | home theater s ( 0.86) | ball pit s ( 0.91) |
| | Supervised ImageNet | auto mechanics indoor s ( 0.96) | N.A. | labyrinth ( 0.94) | N.A. | hay ( 0.94) | N.A. |
| | YFCC-15M | chapel s ( 1) | ice cream parlor s ( 0.99) | pantry s ( 0.97) | temple ( 0.98) | pantry ( 0.97) | temple east asia s ( 0.95) |
| RN-50 | CC-12M | sacristy s ( 0.98) | wheat field s ( 0.98) | funeral chapel s ( 0.97) | meshed ( 0.98) | formal garden s ( 0.96) | polka dotted ( 0.96) |
| | OpenAI | mountain pass s ( 0.99) | knitted ( 0.95) | butte s ( 0.94) | chequered ( 0.91) | water mill s ( 0.89) | wheat field s ( 0.87) |
| | Supervised ImageNet | kiosk indoor s ( 0.95) | N.A. | vegetable garden s ( 0.88) | N.A. | sacristy s ( 0.86) | N.A. |
| | YFCC-15M | liquor store indoor s ( 0.97) | polka dotted ( 0.98) | book stand ( 0.96) | lined ( 0.97) | horse drawn carriage ( 0.89) | dotted ( 0.97) |
| ViT-B/16 | DataComp | carrousel s ( 0.98) | jail cell s ( 0.84) | banquet hall s ( 0.93) | gift shop s ( 0.83) | carport freestanding s ( 0.89) | manhole s ( 0.82) |
| | LAION-2B | flood s ( 0.9) | stained ( 0.89) | catwalk s ( 0.87) | scaly ( 0.86) | rubble ( 0.85) | cracked ( 0.85) |
| | LAION-400M | book stand ( 0.97) | temple ( 0.97) | bookstore ( 0.94) | courtyard s ( 0.95) | rudder ( 0.93) | cabana s ( 0.94) |
| | OpenAI | martial arts gym s ( 0.95) | meshed ( 0.92) | jail cell s ( 0.92) | flecked ( 0.92) | throne room s ( 0.91) | perforated ( 0.92) |
| | Supervised ImageNet | hot tub outdoor s ( 0.99) | N.A. | bedchamber s ( 0.98) | N.A. | stadium baseball s ( 0.97) | N.A. |
| | YFCC-15M | fountain s ( 0.68) | black c ( 0.67) | black c ( 0.67) | N.A. | air base s ( 0.62) | N.A. |
| ViT-B/32 | DataComp | zen garden s ( 0.95) | ice cream parlor s ( 0.67) | dolmen ( 0.94) | bullring ( 0.67) | gift shop s ( 0.88) | junkyard s ( 0.67) |
| | LAION-2B | viaduct ( 0.93) | stained ( 0.91) | cargo container interior s ( 0.91) | scaly ( 0.91) | labyrinth ( 0.9) | matted ( 0.88) |
| | LAION-400M | barnyard s ( 0.86) | scaly ( 0.94) | subway interior s ( 0.86) | jail cell s ( 0.9) | bird feeder ( 0.86) | manhole s ( 0.9) |
| | OpenAI | tennis court s ( 1) | meshed ( 0.95) | batters box s ( 0.98) | perforated ( 0.93) | kennel indoor s ( 0.93) | flecked ( 0.91) |
| | Supervised ImageNet | television studio s ( 0.88) | N.A. | barbecue ( 0.87) | N.A. | lined ( 0.85) | N.A. |

### D.4 POLYSEMANTICTY FOR EACH MODEL

We report the polysemanticity metric computed as per Section 4 for all zero-shot models in Table 10. As claimed in the paper, this ranges from polysemanticity $= 3$ for the OpenAI ResNet 50 to

Table 10: *Polysemanticity of zero-shot models, showing average # of concepts encoded in RSV of last layer (i.e. with* AP $> 0.9$ *on Broden dataset concepts).*

| Backbone | Pretraining data | Polysemanticity |
|---|---|---|
| ResNet50 | OpenAI | 3.0 |
| | YFCC-15M | 6.4 |
| | CC-12M | 5.3 |
| ResNet101 | OpenAI | 3.5 |
| | YFCC-15M | 7.8 |
| ViT-B-16 | OpenAI | 14.5 |
| | LAION-400M | 11.9 |
| | LAION-2B | 16.0 |
| | DataComp | 14.1 |
| ViT-B-32 | OpenAI | 14.1 |
| | LAION-400M | 11.5 |
| | LAION-2B | 11.1 |

polysemanticity $= 16$ for the LAION-2B ViT-B/16, with typically more than 10 concepts encoded in one direction on average.

# E    INTUITION BEHIND GENERALIZATION OF OUTLIER FEATURES

Below, we give more details and an intuition why outlier features can be generalized from the canonical basis $\{e_1, \ldots, e_{d_H}\}$ (we can write $h_i = \text{Proj}_{e_i}(h)$) to be any set of directions of the representation space that receive a projection substantially above average:

Let us assume, for instance, that two of the elements in the canonical basis $e_1$ and $e_2$ correspond to outlier features. This means that an activation vector $h$ related to an input image $x$ has projections $h_1 = \text{Proj}_{e_1}(h)$ and $h_2 = \text{Proj}_{e_2}(h)$ substantially above the average $h_1, h_2 \gg n^{-1} \sum_{i=1}^{n} h_i$. Now let us define a new unit vector $e_1' = 2^{-1/2}(e_1 + e_2)$. We deduce that the projection onto this vector is also substantially higher than average $h_1' = \text{Proj}_{e_1'}(h) = 2^{-1/2}(h_1 + h_2) \gg n^{-1} \sum_{i=1}^{n} h_i$. Hence, the unit vector $e_1'$ can be considered as an outlier feature in a new non-canonical basis. In general, we can extend the notion of *outlier features* to any vector in the span$\{e_1, \ldots, e_n \in \mathbb{R}^n\}$.

# F    PRUNING RESULTS

**Pruning non-privileged directions.** Given that we have established that outlier features induce privileged directions in representation space, it seems interesting to check their role in model performance. To that aim, we gradually prune each RSV $v_i$ by increasing order of $\sigma_i$ by setting $\sigma_i \leftarrow 0$ in the singular value expansion from Equation (3)[4]. By pruning a variable proportion of the singular vectors, we obtain the results in Figure 10. We see that the $80\%$ least important RSV of the representation space can be pruned without a substantial effect on performance, i.e. that the robust models are low-rank in their last layer where they have privileged directions.

However, when extending the pruning experiment to finetuned CLIP models and supervised models trained only with ImageNet, we make the two interesting observations from these new results (see Figure 11):

---

[4]Note that sorting the RSV $v_i$ by increasing $\sigma_i$ is similar to sorting the RSV by increasing $\text{Importance}(i)$, since these two variables are related by a Spearman rank correlation $\rho = 96\%$

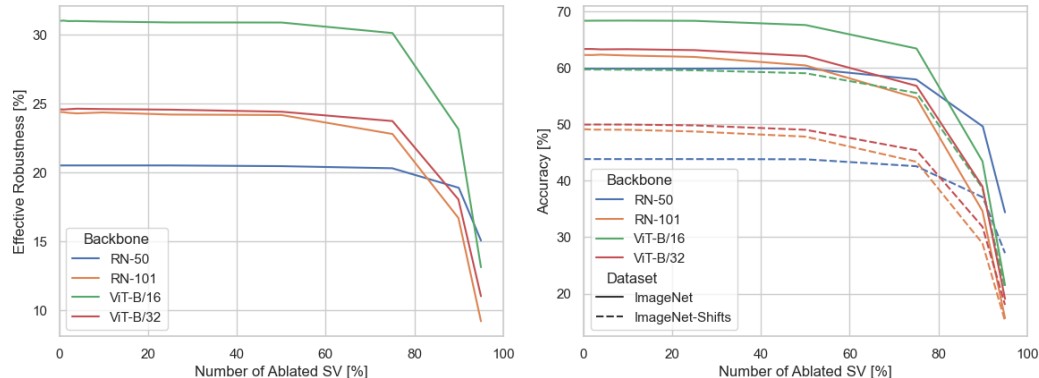

Figure 10: *Effect of gradually pruning the least important SV of W on* ER *and* ACC. The least $80\%$ important SV can be pruned without any substantial effect.

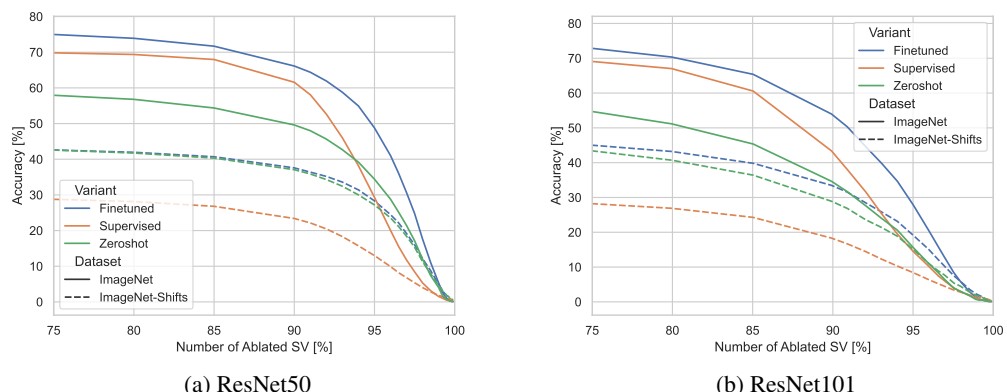

(a) ResNet50                    (b) ResNet101

Figure 11: *Extension of pruning results to finetuned and ImageNet supervised models. Zero-shot models obtained with OpenAI pretraining set.*

**All models are low-rank.** For all the models (zero-shot, finetuned and supervised), the performances are not substantially affected if we remove the $80\%$ least important singular directions of their representation space (compare to Table 4). This shows that many existing models admit good low-rank approximations. This also demonstrates that the fact that these models are low-rank is not necessarily a signature of robustness.

**Faster drop for supervised models.** When the number of ablated singular values ranges between $80\% - 100\%$, we see that the ImageNet accuracy of supervised models drop substantially faster than the accuracy of the finetuned and the zero-shot models. In fact, for the ResNet50, the ImageNet accuracy curves even cross. This implies that the most important direction of the zero-shot model's representation space better discriminate between ImageNet classes than the most important directions of the supervised model's representation space. In the former case, these directions correspond to the zero-shot model's privileged directions. We believe that this new result further reinforces the importance of privileged directions to understand the performances of robust models.

## G    DETAILS ON EXPERIMENTS

### G.1    FINETUNED CLIP MODELS

To obtain the finetuned CLIP models, we proceed as follows. We start from building the zero-shot CLIP models as described in Section 2. As Wortsman et al. (2022b), we then finetune these models

Table 11: *Results of the unique concept analysis, showing total # of unique Broden concepts encoded in last layers along the standard unit vectors $e_i$ of the representation space. Zero-shot models encode substantially more concepts.*

| Backbone | Pretraining data | Zero-shot CLIP *Highest ER* | Finetuned CLIP *Medium ER* |
|---|---|---|---|
| ResNet50 | OpenAI | 395 | 331 |
| ResNet101 | OpenAI | 389 | 331 |
| ViT-B-16 | OpenAI | 648 | 550 |
| ViT-B-32 | OpenAI | 638 | 528 |

for 10 epochs on the ImageNet training set, using a batch size of 256 and a learning rate of $3 \cdot 10^{-5}$ with a cosine annealing learning rate scheduler and a warm-up of $500$ steps. We use the AdamW optimizer and set the weight decay to $0.1$.

### G.2 SUPERVISED IMAGENET MODELS

We note that the ResNets used by Radford et al. (2021) have small modifications, such as the usage of attention pooling. Unfortunately, we are not aware of any public weights for such modified architectures trained on ImageNet from scratch. We thus train these modified ResNet models from scratch for 90 epochs on the ImageNet training set, using a batch size of 1024. We use AdamW, and a learning rate schedule decaying from $10^{-3}$ to $10^{-4}$ after 30 epochs and to $10^{-5}$ after 60 epochs (with a warm-up period of 5,000 steps). We set weight decay to $10^{-2}$. We use the standard augmentations of horizontal flip with random crop as well as label smoothing.

For the ViT models, loadable checkpoints with identical architectures were available from torchvision (TorchVision maintainers and contributors, 2016), and we thus use those directly.

## H CONCEPT PROBING WITH NEURON EXPERTS

In our concept analysis in Section 4, we assign concepts to the directions in representation space that are given as the RSVs of the linear head weight matrix $W$. This was motivated by the fact that we discovered that some of those directions are privileged directions emerging from outlier features for robust models, and wanted to understand what information these directions encode.

However, directions defined by another basis of the representation space can in theory also be used to analyse encoded concepts. In fact, in prior work, individual neurons were used to investigate the concepts encoded by a model (Cuadros et al., 2022), corresponding to the canonical basis of standard unit vectors. We thus replicate the quantitative analysis of Table 3 for the OpenAI zero-shot and finetuned models, but with standard unit vectors $e_i$ replacing $v_i$ in the analysis (for the avoidance of doubt, $e_i$ is defined as a vectors of 0s, but a 1 in its $i$-th entry).

The result is shown in Table 11. Again, we see that the more robust zero-shot models encode substantially more unique concepts than the less robust finetuned models. While we did not have enough time to repeat the analysis for all remaining (including superised) models, these intial results make us confident that our concept analysis of Section 4 also holds if the basis of directions encoding concepts in representation space were changed to individual neuron directions (or any other basis of $\mathbb{R}^{d_H}$ probably).

## I ANALYSIS OF WISE-FT MODELS

In this appendix, we use the approach of Wise-FT (Wortsman et al., 2022b) to obtain a continuous spectrum of ER. Given a zero-shot model $f_{\theta_0}$ with weights $\theta_0 \in \Theta$ and a finetuned model $f_{\theta_1}$ with weights $\theta_1 \in \Theta$, Wortsman et al. (2022b) propose to interpolate between the two models in weight space. This is done by taking a combination $\theta_\alpha := (1 - \alpha) \cdot \theta_0 + \alpha \cdot \theta_1$ for some interpolation parameter $\alpha \in [0, 1]$. One then defines a new model $f_{\theta_\alpha}$ based on the interpolated weights.

Surprisingly, interpolating between zero-shot CLIP models and finetuned CLIP models produce models with good performances. To illustrate that, we perform the Wise-FT interpolation with all

the models from our pool. We report the ImageNet & shift accuracies of these models in Figures 12 and 13. For the OpenAI and LAION models in Figure 12, we observe that the shift accuracy of interpolated models often surpass both the zero-shot and the finetuned models. The YFCC-15M and CC-12M models in Figure 13 exhibit a different trend: both ImageNet & shift accuracies increase monotonically as $\alpha$ sweeps from zero-shot to finetuned. This is likely due to the low accuracy of the corresponding zero-shot models.

By analyzing the ER of interpolated OpenAI and LAION models in Figure 14, we see that the ER gradually degrades as $\alpha$ sweeps between the zero-shot and the finetuned models. Interestingly, the ER of YFCC-15M and CC-12M models in Figure 15 peaks at $\alpha = .4$ and then decreases monotonically.

Let us now look at how our ER signatures evolve as we sweep $\alpha$ between zero-shot and finetuned models. Ideally, if these signatures are good ER proxies, they should exhibit similar trends as the ones described in the previous paragraph. For the OpenAI and LAION models, we indeed observe in Figures 16 and 18 that the kurtosis and the number of unique encoded concepts gradually decrease as $\alpha$ sweeps from zero-shot to finetuned models. Similarly, we observe in Figures 17 and 19 that these two metrics start to substantially after $\alpha = 0.4$ for the YFCC-15M and CC-12M models. This suggests that these two metrics constitute a good proxy to track how the ER of a given model evolves.

Note that the Wise-FT idea has since been generalized to a combination of several finetuned models by Wortsman et al. (2022a) with *model soups*. We leave the investigations of model soups for future work.

## J    FURTHER LITERATURE

**Defining CLIP ER.** The definition of ER crucially relies on the observation in multiple works that the model performance on natural shifts is linearly related to its performance in-distribution when both quantities are plotted with a logit scaling (Recht et al., 2018; 2019; Miller et al., 2020). We note though that there are known exceptions to this, e.g. considering out-of-distribution generalization on real-world datasets that substantially differ from the in-distribution dataset (Fang et al., 2023).

**Explaining CLIP ER.** A first intuitive explanation for the surprisingly high effective robustness of CLIP might be the fact that the learned embeddings are endowed with semantic grounding through pretraining with text data. This hypothesis was refuted by Devillers et al. (2021), who demonstrated that the embeddings in CLIP do not offer gains in unsupervised clustering, few-shot learning, transfer learning and adversarial robustness as compared to vision-only models. In a subsequent work, Fang et al. (2022) demonstrated that the high robustness of these models rather emerges from the high diversity of data present in their training set. This was achieved by showing that pretraining SimCLR models Chen et al. (2020) on larger datasets, such as the YFCC dataset by Radford et al. (2021), *without any language supervision* matches the effective robustness of CLIP. Shi et al. (2023) reinforced this data-centric explanation by showing that the performance on the pretraining set also correlates linearly with the out-of-distribution performance. To put the emphasis on the importance of data-quality for effective robustness, Nguyen et al. (2022) showed that increasing the pretraining set size does not necessarily improve the effective robustness of the resulting model. Rather, it suggests that it is preferable to filter data to keep salient examples, as was done, e.g., to assemble the LAION dataset (Schuhmann et al., 2022).

**Other signatures of ER.** By comparing pretrained models with models trained from scratch, Neyshabur et al. (2020) demonstrated that these models exhibit interesting differences, such as their reliance on high-level statistics of their input features and the fact that they tend to be separated by performance barriers in parameter space. Guillory et al. (2021) found observable model behaviours that are predictive of effective robustness. In particular, the difference of model's average confidence between the in and out-of-distribution correlates with out-of-distribution performance.

**Polysemanticity in foundation models.** Polysemantic neurons were coined by Olah et al. (2020) in the context vision model interpretability. These neurons get activated in the presence of several unrelated concepts. For instance, the InceptionV1 model has a neurons that fires when either cats or cars appear in the image. These neurons render the interpretation of the model substantially more complicated, as they prevent to attach unambiguous labels to all the neurons in a model. This will

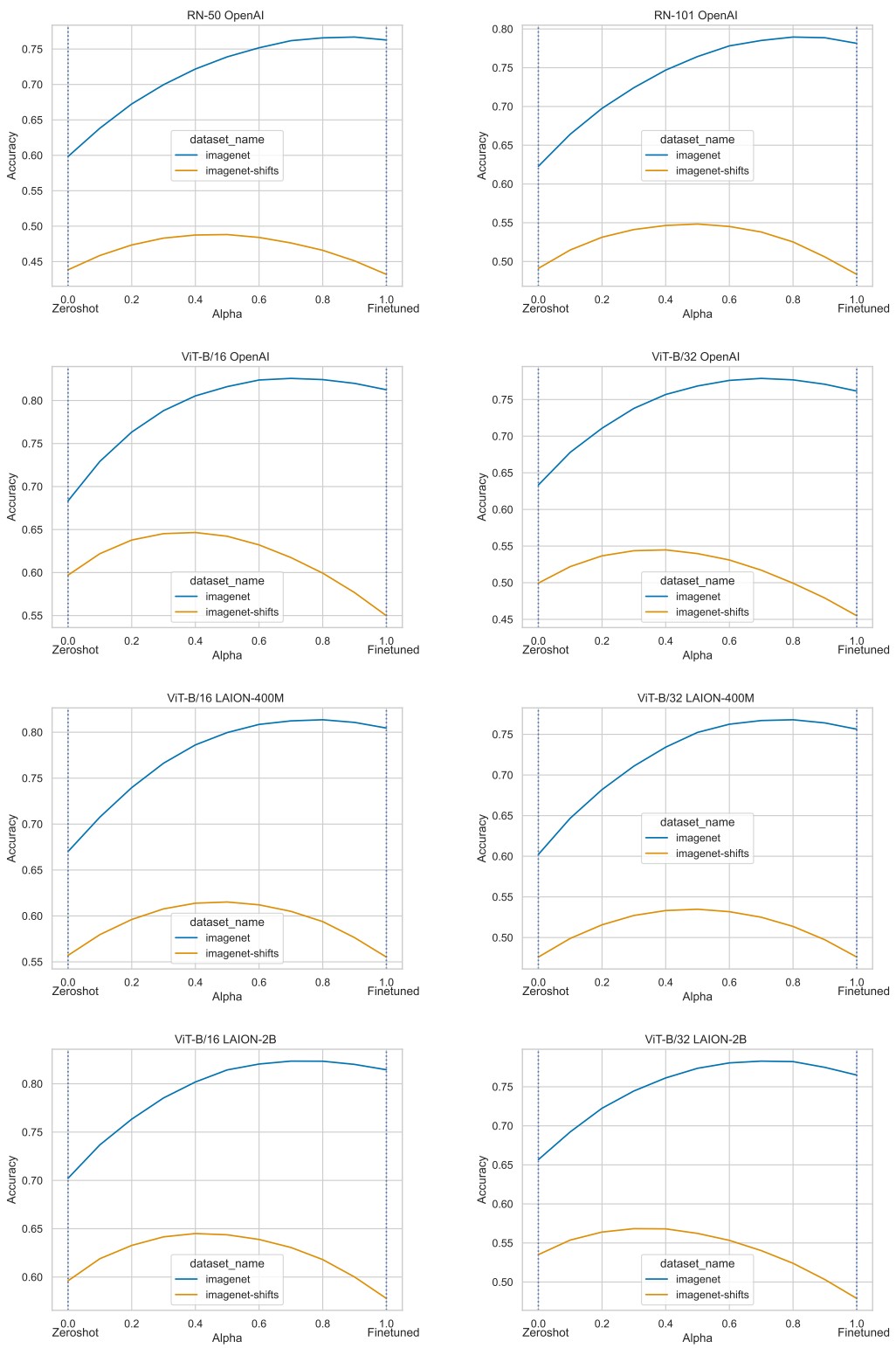

Figure 12: *Accuracies on ImageNet & shifts for Wise-FT models (1/2).*

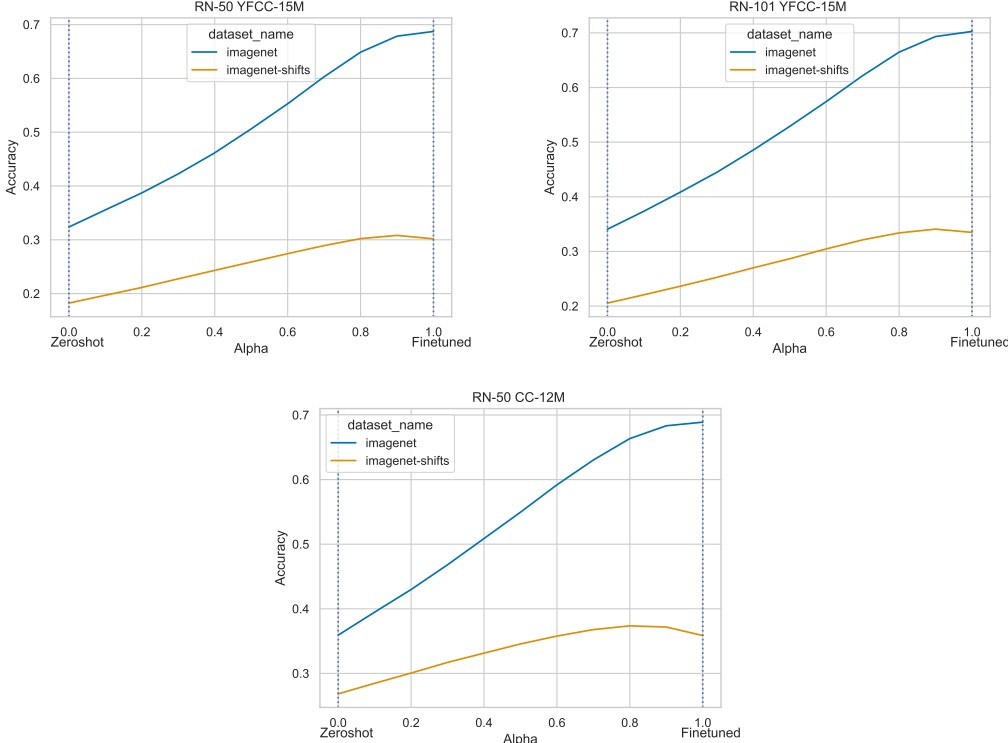

Figure 13: *Accuracies on ImageNet & shifts for Wise-FT models (2/2).*

limit the insights gained by traditional interpretability techniques. For example, producing saliency maps for a polysemantic neuron could highlight many unrelated parts of an image, corresponding to the unrelated concepts this neuron is sensitive to. A qualitative analysis of the neurons in CLIP by Goh et al. (2021) showed that a CLIP ResNet has a substantial amount of polysemantic neurons. The emergence of polysemantic neurons is a complex phenomenon. It is not yet well-understood for models at scale. The latest works on the subject mostly focus on toy models, see e.g. the works of Elhage et al. (2022) and Scherlis et al. (2022). To the best of our knowledge, our work is the first to explicitly discusses the link that exists between polysemanticity and robustness to natural distribution shifts.

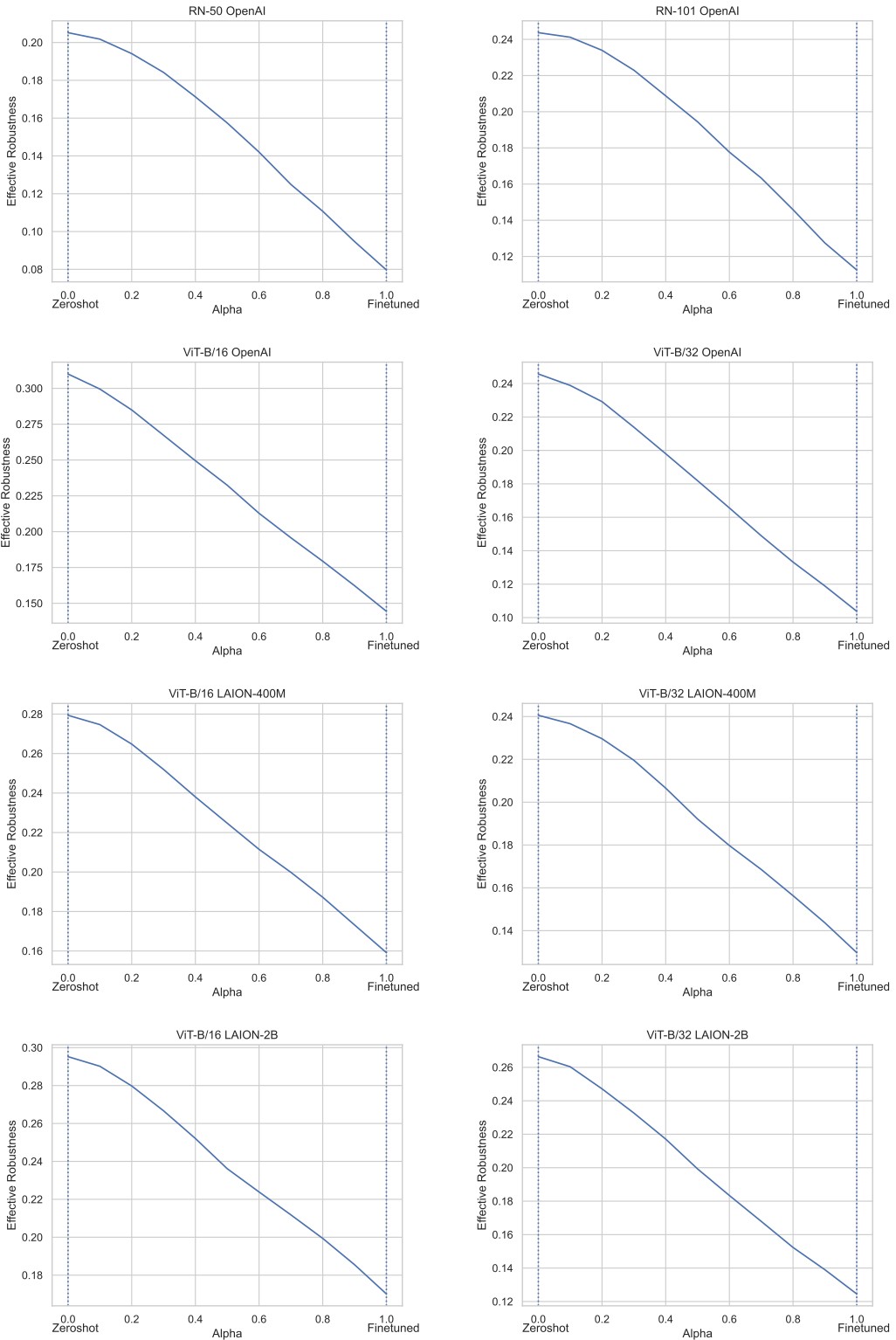

Figure 14: *ER for Wise-FT models (1/2).*

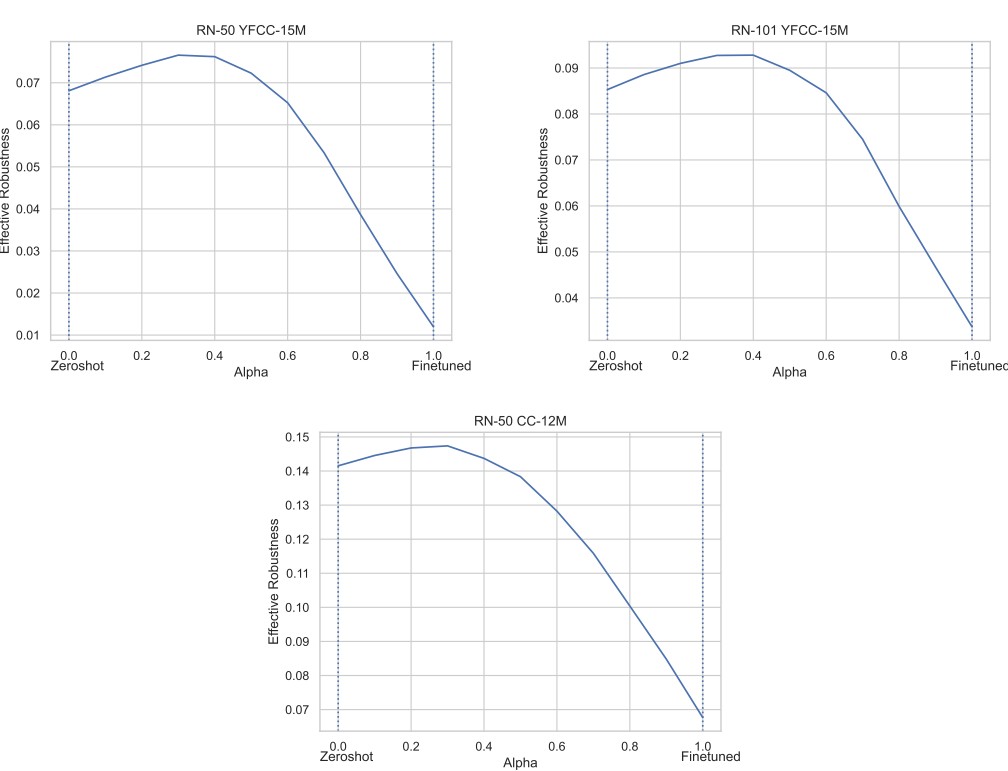

Figure 15: *ER for Wise-FT models (2/2).*

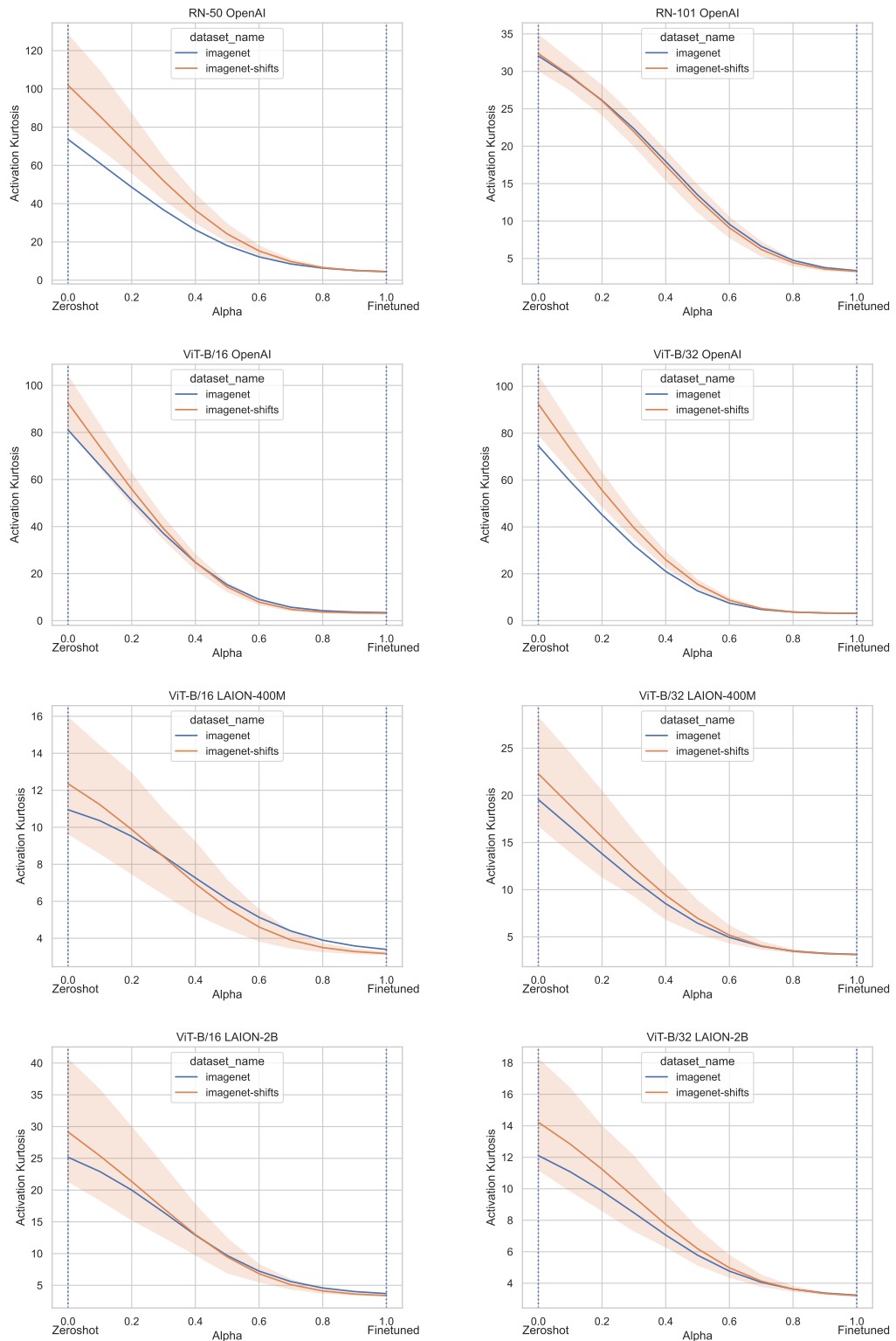

Figure 16: *Activation kurtosis for Wise-FT models (1/2). The activation kurtosis is computed for both ImageNet tests and ImageNet shifts.*

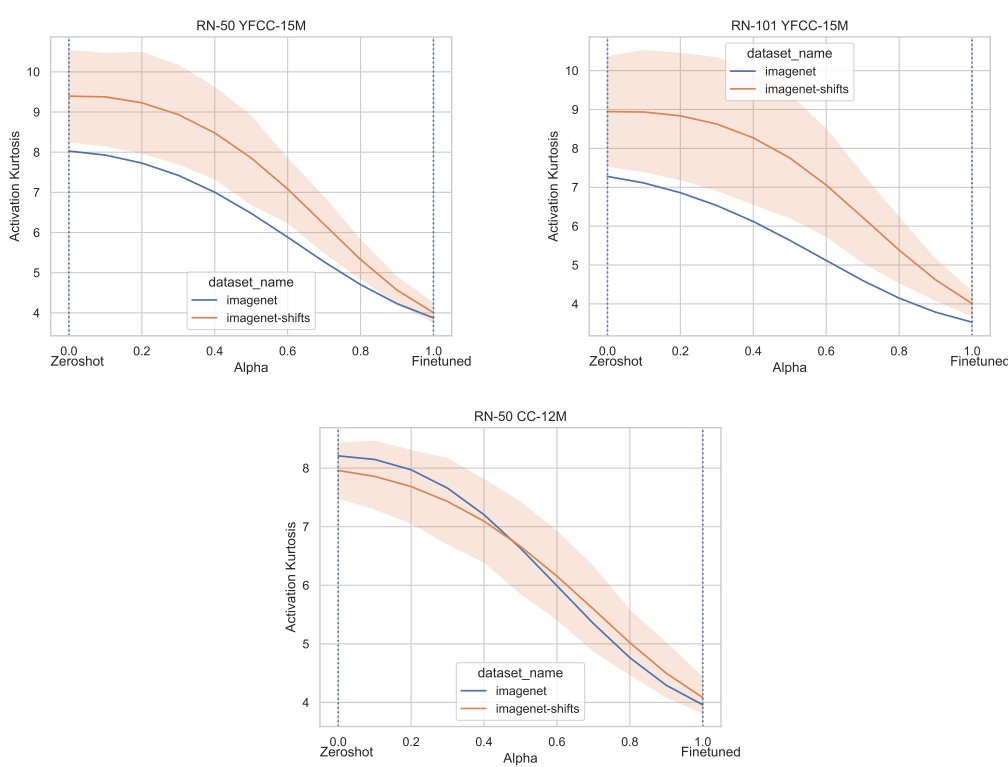

Figure 17: *Activation kurtosis for Wise-FT models (2/2). The activation kurtosis is computed for both ImageNet tests and ImageNet shifts.*

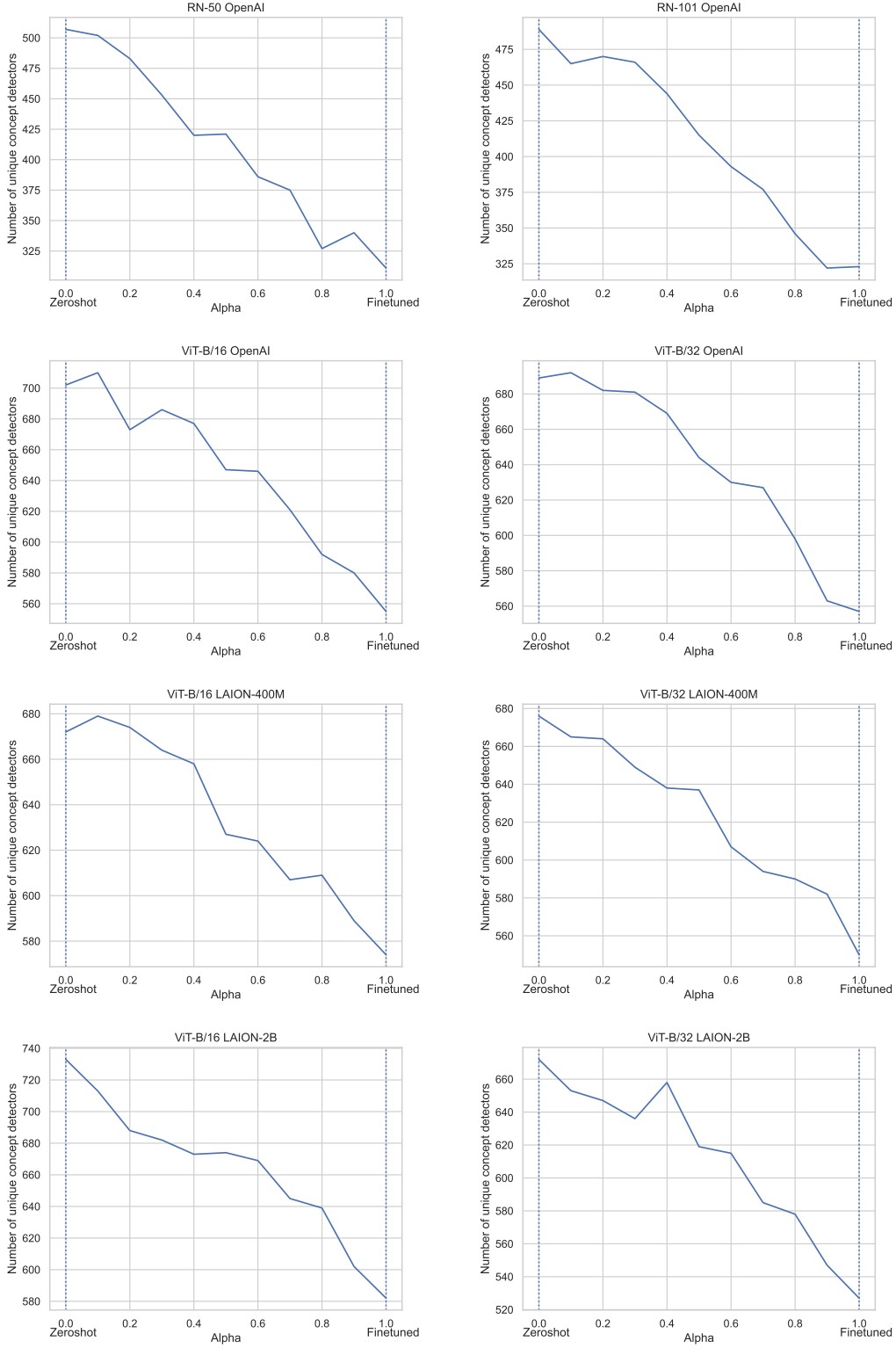

Figure 18: *Number of unique concepts encoded in Wise-FT models (1/2).*

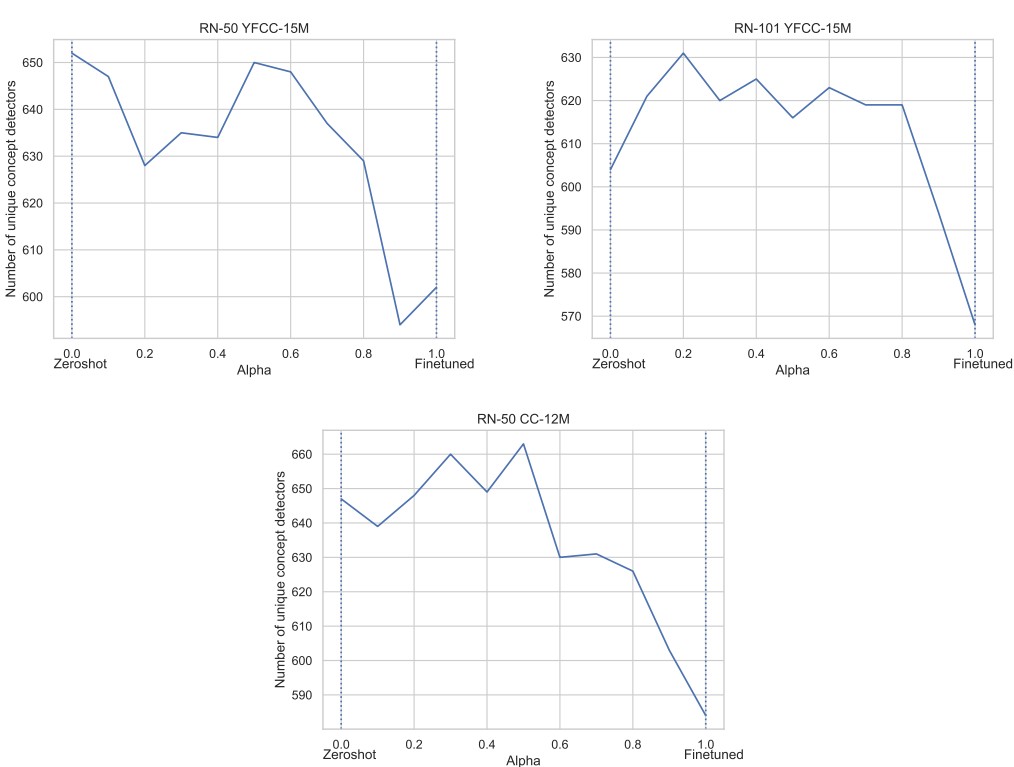

Figure 19: *Number of unique concepts encoded in Wise-FT models (2/2).*

