# OpenReview forum: "Robust multimodal models have outlier features and encode more concepts"
_ICLR.cc/2024/Conference — Submitted to ICLR 2024_

### Official Review · Reviewer_8h4M · 2023-10-29

**Soundness:** 1 poor
**Presentation:** 3 good
**Contribution:** 2 fair
**Rating:** 5
**Confidence:** 3

**Summary:**

The goal of this paper is to characterize CLIP-style models in terms of their weights and hidden representations. This is motivated by these model’s superior generalization ability, which suggests that their internal representations are qualitatively different than smaller models trained on less data. The work finds that robust models exhibit outlier features, that is, individual neuron activations with significantly higher magnitude than the average. The work also tries to quantify the number of distinct concepts that these models learn (relative to models either finetuned or trained from scratch on ImageNet), with results suggesting that models with CLIP-training on large datasets learn more concepts.

**Strengths:**

- **Motivation:** As greater and greater resources are poured into training frontier models, developing an understanding of what makes these models so much more robust than their counterparts trained on smaller datasets becomes a more and more pressing problem. The approach taken by this paper, attempting to characterize robustness, through model structure alone, is promising as a low compute way of assessing the robustness of a model and a starting point to finding a mathematical basis for robustness.
- **Clarity:** The work is mostly well-written and well-structured. The reviewer particularly appreciated the way that the work summarizes key takeaways in colored boxes that are easy to find. The visualizations and plots were well-thought-out and communicated their information efficiently.
- **A strong premise:** Though the reviewer had some concerns around the way the experiments were run and the conclusions that were drawn from them (see the "Weaknesses" section), the premise is very interesting and feels like a strong avenue for further exploration.

**Weaknesses:**

- **Experiments:** From this reviewer’s perspective, the main issues with the paper come from the experiments and the conclusions drawn from these. We outline our concerns below.
    1. **Privileged directions experiment:** This reviewer did not understand how the privileged directions experiment connected with outlier features. To check our understanding, outlier features are a subset of activations that have substantially higher magnitude than the average activation for that input. Privileged directions on the other hand, are (a subset of?) the right singular vectors for the final linear layer. To determine whether a right singular vector $v_i$ is important to the encoder, the cosine similarity between $v_i$ and a sequence of activation vectors is computed and scaled by the corresponding singular value. This is termed the *importance* of $v_i$. It is not clear to the reviewer how ‘importance’ says anything about outlier features. The latter seems to be a property specifically related to the activation (or neuron) basis whereas the former consists of right singular vectors of the weight matrix, which are almost certainly not the activation basis. To be clear, identifying whether activations align with singular vectors with large singular value seems interesting, it just doesn’t seem to be related to outlier features. Perhaps the tool one wants is closer to a measure of sparsity?
    2. **Pruning experiment:** In Section 3, “Pruning non-privileged directions”, the paper notes that one can prune away the smallest 80% of all singular vectors without substantial loss of accuracy. This technique is a standard method of finding a low-rank approximation of a matrix and well-studied. For large matrices, it does not seem surprising that the impact on classification is small even when using a low-rank approximation as the data itself can be approximated by a low-dimensional subspace of the ambient space. Furthermore, it is hard to tell if the results are specific to robust models when this technique is not applied to the finetuned and trained from scratch models.
- **Lack of coverage of the multimodal aspect of the models:** Given the title, one would expect that there would be some discussion about how the multimodal aspect of CLIP-type models impacts the results. Surprisingly, this feature was never addressed. This reviewer would suggest either removing the word ‘multimodal’ from the title or adding a section to address this aspect. Further, it would be more precise to say that the work considers CLIP-type models since the experiments focus exclusively on this particular family.
- **Experimental breadth:** Related to the previous point, it would make the work stronger if the results were expanded, either by increasing the breadth of the experiments (more models for instance) or providing some analysis of why we see the phenomena that we do.

### Nitpicks
- The abstract and introduction use the term *outlier feature* without any explanation. The term is not defined for several pages. While outlier features are known within the interpretability community, for the sake of accessibility, this reviewer would recommend putting at least an informal explanation of what this concept is the first time it is mentioned.
- It’s possible the reviewer missed it, but it seems that $d_X$ and $d_H$ are never defined in Section 3, “Approach”.
- SVD is a foundational method in linear algebra. As such, there probably isn’t a reason to include (3).
- The reviewer appreciates the validation of previous work showing trends in effective robustness (ER). It would be good if more papers did these kinds of validation experiments. On the other hand, as space is precious, this reviewer would suggest moving some of the text in this section to the appendix to focus on this work’s contributions. It would seem that the main point the paper needs to make with regard to ER is that CLIP-style models stand-out for their ER relative to the same architectures finetuned on ImageNet or trained from scratch on ImageNet. This could be done more concisely.

**Questions:**

- Looking at Table 3, one sees that the ImageNet supervised models tend to often have more concepts than some of the finetuned CLIP models. Is there an explanation for why this is?
- Figure 4 suggests that the ViT models tend to share more of the same concepts between training techniques, are there any guesses for why this is?
- This reviewer did not understand the remark “An interesting parallel can be drawn with the work of Bondarenko et al. (2023), which found that outlier features in language models assign most of their mass to separator tokens (such as the end of sentence token).”

---

> ### Author Response · Authors · 2023-11-18
> **[1/3] Rebuttal to Reviewer 8h4M**
>
> We thank the reviewer for their thoughtful comments and suggestions. Below, we address all the reviewer's remarks. We are keen to address any remaining concerns the reviewer might have during the discussion period.
>
>
> ## Re Weakness 1.1 (Priviledged directions and outlier features)
>
>
> First, we would link to confirm that the understanding of the reviewer is correct: priviledged directions are a subset of the right singular vectors of the final weight matrix.
>
> We appreciate the opportunity to elaborate on a crucial point: the connection between privileged directions, outlier features, high kurtosis.
>
> Privileged directions are one of our contributions and denote outlier features that not only receive substantially higher activations but also contribute significantly to the logits.
>
> Outlier features are directions that receive a substantially higher activation, this term was introduced in  **[1]**. Compared to Privileged Direction, outliers features may or may not contribute significantly to the logits.
>
> A high kurtosis signals that some directions of the model’s representation space receive a substantially higher activation than the other directions. However, since the kurtosis metric defined in *Equation (2)* aggregates over all components of the activation vectors, it is not possible to identifies which directions of the representation space are outlier features.
>
> **Summary**. In summary, high kurtosis indicates the presence of outlier features, without identifying them. Outlier features represent directions with high activations, regardless of their contribution to the logits. Privileged directions are a subset of outlier features that also contribute substantially to the logits and consequently, to the final predictions.
>
>
> Below we offer a more formal explaination where we also explain how we identify priviledged directions using the importance score.
>
> **Large activations in some directions.** Next, we would like to clarify the construction of the importance scores and how these relate to outlier features. Consider a basis $\{ v_i \in \mathbb{R}^{d_H} \mid i \in [d_H] \}$ of the representation space $\mathbb{R}^{d_H}$, where $d_H \in \mathbb{N}^+$ is the dimension of the representation space. Let us assume that the direction $v_j$ receive substantially higher activations, hence corresponding to an outlier feature. This implies that the absolute cosine similarity of an activation vector $h^{(n)} \in \mathbb{R}^{d_H}$ with this direction will typically be higher than average: $|\cos (v_j, h^{(n)})| \gg \frac{1}{d_H} \sum_{i=1}^{d_H} |\cos (v_i, h^{(n)})|$. This corresponds to the second factor in our importance score $\mathrm{Importance}(j)$ defined in *Equation (4)*.
>
> **Contribution to logits.** Now what guarantees that this outlier feature $v_j$ is indeed important for the model? As we argue in the paper, this direction of the latent space needs to play a role in the computation of the logits. In order to restrict to directions of the latent space that matter for the classifier, we study the directions of the latent space corresponding to right singular vectors $\{ v_i \in \mathbb{R}^{d_H} \mid i \in [\mathrm{rank}(W)] \}$ of the classification head $W \in \mathbb{R}^{d_Y \times d_H}$, where $d_Y = 1,000$ is the number of ImageNet classes. Indeed, this set of vector spans the subspace of $\mathbb{R}^{d_H}$ that is orthogonal to the kernel of the classification head $\mathrm{span} \{ v_i \in \mathbb{R}^{d_H} \mid i \in [\mathrm{rank}(W)] \} = \ker(W)^{\perp}$, which means that they allow us to decompose any activation vector $h \in \mathbb{R}^{d_H}$ whose contribution to the logits is nonzero. Now the outlier feature $v_j$ has a substantial contribution to the logit only if it corresponds to a large singular value $\sigma_j \gg 0$. This motivates our characterization of *priviledged directions* as directions of the representation space that receive a large activation (and, hence, correspond to outlier features) *and* that have a substantial contributions to the logits. This extra contribution corresponds to the first factor in our importance score $\mathrm{Importance}(j) \in \mathbb{R}^+$ defined in *Equation (4)*. Priviledged directions are then characterized by an importance score substantially higher than average $\mathrm{Importance}(j) \gg \frac{1}{d_H} \sum_{i=1}^{\mathrm{rank} (W)} \mathrm{Importance}(i)$.
>
> **[1]** Dettmers, T., Lewis, M., Belkada, Y., & Zettlemoyer, L. (2022). Llm. int8 (): 8-bit matrix multiplication for transformers at scale. arXiv preprint arXiv:2208.07339.

---

> ### Author Response · Authors · 2023-11-18
> **[2/3] Rebuttal to Reviewer 8h4M**
>
> ## Re Weakness 1.2 (Pruning experiment)
>
> We thank the reviewer for giving some further context to our pruning experiment. We agree with the reviewer that many works exist on studying low-rank approximation of neural networks. We would like to clarify that we do not claim that this low-rank approximation is a signature of robustness. Rather, we claim that for zeroshot CLIP models, this confirms the fact that a small number of latent space directions (including these priviledged directions) explain the high performances of the model, and hence, its high effective robustness.
>
> Additionally, following the reviewer's recommendation, we have extended the pruning experiment from *Section 3* to finetuned CLIP models and supervised models trained only with ImageNet. These new results are hosted at [this link](https://imgur.com/a/QWuahzn). We make two interesting observations from these new results.
>
> **All models are low-rank.** For all the models (zeroshot, finetuned & supervised), the performances are not substantially affected if we remove the $80 \\%$ least important singular directions of their representation space. This confirms the reviewer's point that many existing models admit good low-rank approximations. This also confirms that the fact that these models are low-rank is not necessarily a signature of robustness.
>
> **Faster drop for supervised models.** When the number of ablated singular values ranges between $80\\%-100\\%$, we see that the ImageNet accuracy of supervised models drop substantially faster than the accuracy of the finetuned and the zeroshot models. In fact, for the ResNet50, the ImageNet accuracy curves even cross. This implies that the most important direction of the zeroshot model's representation space better discriminate between ImageNet classes than the most important directions of the supervised model's representation space. In the former case, these directions correspond to the zeroshot model's priviledged directions. We believe that this new result further reinforces the importance of priviledged directions to understand the performances of robust models.
>
>
> These new results make a great addition to the manuscript. Hence, we have added the above discussion in *Appendix D* of the updated manuscript.
>
> ___
> ## Re Weakness 2. (Multimodal aspects)
>
>
> We agree with the reviewer that our study mostly focuses on CLIP-style models. After an internal discussion and to avoid any confusion, we have decided to update the title to:
>
> *Robustness Signatures in CLIP Models*
>
> It is also correct that our work restricts to the vision encoder of CLIP models. This is because CLIP vision encoder is very often used without the text encoder in practice. For instance, DALL-E 3 only leverages the vision encoder of CLIP and throws away the text encoder, see e.g. [this video](https://www.youtube.com/watch?v=pgaTOX-RUQ4) by the authors of DALL-E. Another example more related to NLP is BLIP-2 **[1]**, which leverages CLIP vision encoder without its language encoder.
>
> **[1]** Li, J., Li, D., Savarese, S., & Hoi, S. (2023). Blip-2: Bootstrapping language-image pre-training with frozen image encoders and large language models. arXiv preprint arXiv:2301.12597.
> ___
> ## Re Questions 1.-3. (Remarks on the concept analysis)
>
> The reviewers raised several interesting questions with respect to our concept analysis. Below, we answer each of those.
>
> **Q1: More concepts in supervised models.** As the reviewer pointed out, supervised models often have more concepts encoder than finetuned models. We hypothesize that this is due to the fact that finetuning is done for a few epochs (only 10), hence making the models forget many zeroshot concepts without giving it the time to memorize all the ImageNet concepts. The concept forgetting is illustrated by the Venn diagrams in *Figure 4*, where we observe that the finetuned concepts are almost subsets of the zeroshot concepts.
>
> **Q2: Better concept alignment between ViTs.** We thank the reviewer for pointing out this interesting observation! Unfortunately, at this point, we also cannot offer an explanation as to why this happens. If the reviewer has a specific analysis in mind that could help understand this better, we would be keen to look into it.
>
> **Q3: Parallel with the work of Bondarenko et al.** We would like to clarify the parallel described in the paragraph *Interpreting priviledged directions* of *Section 4*. As shown in our analysis, the concepts encoded in the outlier features represent regular alternating patterns (like presence/absence of holes). The outlier features in LLMs focus their attention to separator tokens, which appear regularly and alternate with more informative tokens. Hence, a parallel can be made by considering the visual alternating patterns (like holes) to mimic the behaviour of text separator tokens. Since this parallel is not a key contribution of our work but rather an interesting curiosity, would this reviewer recommend to move this to the Appendix?

---

> ### Author Response · Authors · 2023-11-18
> **[3/3] Rebuttal to Reviewer 8h4M**
>
> ## Re Weakness 3. (Experimental breadth)
>
> We thank the reviewer for their suggestion. Following their recommendation, we have increased the breadth of our experiments in two meaningful ways: by adding non-CLIP style models (CoCa **[1]**), as well as by included larger scale models (with ViT-L-14 backbones). All these results have been added to *Appendix C.1* of the updated manuscript.
>
> Below, we verify that the empirical results from our paper extend to these models. First we confirm that all these models have high effective robustness when used as zero-shot classifiers. As in the paper, we observe that finetuning on ImageNet decreases the effective robustness of these classifiers.
>
> |**Backbone**|**Pretraining Data**|**Zero-shot ER**|**Finetuned ER**|
> |---|---|---|---|
> |COCA ViT-B-32|LAION-2B|25%|14%|
> |COCA ViT-L-14|LAION-2B|34%|21%|
> |CLIP ViT-L-14|OpenAI| 37% | 21%|
> |CLIP ViT-L-14|LAION-400M|32%|20%|
> |CLIP ViT-L-14|LAION-2B|32%|21%|
> |CLIP ViT-L-14|DataComp|37%|24%|
> ||
>
> Next, we show that all the zero-shot models have high kurtosis, which implies the existence of outlier features in their representation space. Additionally, we show that finetuning again decreases the kurtosis.
>
> |**Backbone**|**Pretraining Data**|**Zero-shot kurtosis**|**Finetuned kurtosis**|
> |---|---|---|---|
> |COCA ViT-B-32|LAION-2B|12.0|3.6|
> |COCA ViT-L-14|LAION-2B|15.5|4.6|
> |CLIP ViT-L-14|OpenAI| 60.8 | 4.6|
> |CLIP ViT-L-14|LAION-400M|20.3|5.2|
> |CLIP ViT-L-14|LAION-2B|66.2|6.9|
> |CLIP ViT-L-14|DataComp|37.4|4.6|
>
>
>
>
> Finally, we check that zero-shot model encodes more concepts. Again, we see that finetuning removes some concepts from the model's representation space.
> |**Backbone**|**Pretraining Data**|**Zero-shot #concept**|**Finetuned #concept**|
> |---|---|---|---|
> |COCA ViT-B-32|LAION-2B|674|530|
> |COCA ViT-L-14|LAION-2B|747|629|
> |CLIP ViT-L-14|OpenAI| 704 | 623|
> |CLIP ViT-L-14|LAION-400M|683|613|
> |CLIP ViT-L-14|LAION-2B|704|633|
> |CLIP ViT-L-14|DataComp|684|619|
>
>
> **[1]** Jiahui Yu, Zirui Wang, Vijay Vasudevan, Legg Yeung, Mojtaba Seyedhosseini, Yonghui Wu. CoCa: Contrastive Captioners are Image-Text Foundation Models

---

> ### Author Response · Authors · 2023-11-21
> **Follow-up to the Rebuttal for Reviewer 8h4M**
>
> We would like to follow-up on the **Q2: Better concept alignment between ViTs** paragraph in Part [2/3] of our rebuttal.
>
> We manually inspected the concepts encoded by the models shown in *Figure 4* to investigate this further. We found a number of concepts that lie in the intersection of all models (grey area) for the ViT models, but were only encoded in the ResNets that were trained on ImageNet (through finetuning or supervised from scratch), i.e. that lie in the green, teal, or purple-blue areas. These concepts are:
>
> Baptistry-indoor-s, barbershop-s, castle-s, courtyard-s, crate, gas-station-s, home-theater-s, hunting-lodge-s, pantry, ranch-house-s, reading-room-s, skittle-alley, town_house-s, watchtower.
> Furthermore concepts related to pool tables (corner pocket, pool table, poolroom-establishment-s, poolrome-home-s) and elevators (elevator-lobby-s, elevator_shaft-s, elevator-door-s, elevator-freight_elevator-s, elevator-interior-s).
>
> These concepts are all related to objects in scenes. It seems like while ViTs picked them up during pretraining (hence also the zeroshot model encodes them), ResNets only learn to encode them when training on ImageNet, where they seem to be more important.
> ___
> We again thank the reviewer for their useful comments.
> We hope that our rebuttal has addressed any remaining concern about the paper.
> If not, we would like to kindly ask the reviewer to engage in discussion before the end of the discussion period (tomorrow).
> Otherwise, we hope that the reviewer will consider updating their recommendation accordingly.

---

> > ### Comment · Reviewer_8h4M · 2023-11-23
> > **Replies to rebuttal**
> >
> > We would like to thank the authors for clarifying a number of aspects of their work and for describing a substantial number of new results. The increased experimental breadth and the further investigation into pruning are particularly appreciated.
> >
> > The remaining confusion that the reviewer has can be reduced to two sentences from the rebuttal above:
> >
> > 1. "priviledged directions are a subset of the right singular vectors of the final weight matrix".
> > 2. "Privileged directions are one of our contributions and denote outlier features that not only receive substantially higher activations but also contribute significantly to the logits."
> >
> > It is our understanding from the descriptions in this work and others that outlier features are specific coordinates (in the neuron basis) in model hidden representations with substantially larger magnitude. Thus, geometrically we can think of them as elements of the elementary unit vectors $e_1, e_2, \dots, e_n \in \mathbb{R}^n$, where $\mathbb{R}^n$ is the particular hidden representation space. On the other hand, the right singular vectors cannot in general be expected to belong to $e_1, e_2, \dots, e_n$. Is there something special in this situation that the reviewer is missing?

---

> ### Author Response · Authors · 2023-11-23
> **Re Follow up Questions (Outlier Feature vs. Priviledged Direction)**
>
> We thank the reviewer for narrowing down the confusion. We agree that this point deserves further clarifications. It is correct that previous work considered *outlier features* to be a subset of vectors in the canonical basis $\\{ e_1, \dots, e_n \in \mathbb{R}^n \\}$ of the $n$-dimensional representation space. With the introduction of privileged directions, which are right singular vectors of the weight matrix $W$, we generalize this definition to *any set of directions of the representation space that receive a projection substantially above average*. We explain this in more detail below, and have also added a clarification in the manuscript (see *Section 3* of the revised manuscript).
>
> Let us imagine, for instance, that two of the elements in the canonical basis $e_1$ and $e_2$ correspond to outlier features. This means that an activation vector $h$ related to an input image $x$ has projections $h_1 = \mathrm{Proj}\_{e_1}(h)$ and $h_2 = \mathrm{Proj}\_{e\_2}(h)$ substantially above the average $h_1, h_2 \gg n^{-1} \sum_{i=1}^n h_i$. Now let us define a new unit vector $e_1' = 2^{-1/2} (e_1 + e_2)$. We deduce that the projection onto this vector is also substantially higher than average $h_1' = \mathrm{Proj}\_{e_1'}(h) = 2^{-1/2} (h_1 + h_2) \gg n^{-1} \sum_{i=1}^n h_i$. Hence, the unit vector $e_1'$ can be considered as an outlier feature in a new non-canonical basis. In general, we extend the notion of *outlier features* to any vector in the $\mathrm{span} \\{ e_1, \dots, e_n \in \mathbb{R}^n \\}$. This includes the right singular vectors of the weight matrix $W$, i.e. our privileged directions.
>
>
> We acknowledge that this relaxed definition of *outlier features* distinguishes our work from the related literature, and deserves some further clarifications. We thank the reviewer for pointing this out, and have clarified this point in *Section 3* of the revised manuscript. We thank again the reviewer for their engagement, and hope this clarifies any residual ambiguity.

---

### Official Review · Reviewer_YHoY · 2023-10-30

**Soundness:** 4 excellent
**Presentation:** 3 good
**Contribution:** 3 good
**Rating:** 8
**Confidence:** 4

**Summary:**

This paper empirically showed that robust multimodal models have outlier features and these outlier features encode more concepts. The paper analyzed the representation spaces of various multimodal models and found that more robust models have much more outlier features. What’s more, by probing these outlier features of robust multimodal models, the authors find that the principled directions in them encode substantially more concepts.

**Strengths:**

1. The paper is tackling an important problem in understanding robust multimodal models. Multimodal models are often found to be more robust than prior supervised models. It is not clear why that is the case. This paper provides many intriguing evidences for this observation.
2. The finding that robust multi-modals have more outlier features is interesting. It is another evidence that zero-shot CLIP models are much different than other non-robust models.
3. The analysis of the paper is thorough, including using activation kurtosis to analzye outlier features and also the use of concept probing.

**Weaknesses:**

1. It is not clear to the me, what is the reason for selecting the metric of activation kurtosis for the analysis in Section 3. What makes this metric interesting for the analysis of outlier features?
2. It seems section 2 is an re-evaluation of existing works on effective robustness. It would be good to summarize these results and definitions in a concise fashion.

**Questions:**

1. In table 2, why would the kurtosis of OpenAI CLIP models be much higher? It seems to be an extreme value. I am interested as to what would be the difference between OpenAI and YFCC-15M/CC-12M CLIP models.
2. From equation 3, the definition of privileged directions in representation space seems to be based on SVD decomposition of the classification head. Have the authors tried more involved methods, e.g. reduced rank regression? Instead of finding the low-rank approximation of W, reduced rank regression would find the low-rank approximation of WX.

---

> ### Author Response · Authors · 2023-11-18
> **[1/2] Rebuttal to Reviewer YHoY**
>
> We thank the reviewer for their thoughtful comments and suggestions. Below, we address all the reviewer's remarks. We are keen to address any remaining concerns the reviewer might have during the discussion period.
>
>
> ## Re Weakness 1. (Activation kurtosis)
>
>
> Our motivations for choosing the activation kurtosis as a first metric to detect outlier features is similar to **[1]**. The idea is that outlier features create heavy tails in the activation distributions, which is detected by the kurtosis metric.We shall detail this argument below for completeness.
>
> Let us first imagine that a representation space has no priviledged direction. This implies that the model's features are represented in random directions. It turns out that random unit vectors can be obtained by normalizing samples from an isotropic Gaussian. Therefore, if the representation space admits no priviledged direction, we expect the distribution of activation vectors $\{ h^{(n)} \in \mathbb{R}^{d_H} \mid n \in [N]\}$, where $N$ is the number of instances for which the activation vector is computed and $d_H \in \mathbb{N}^+$ is the dimension of the representation space, to be well-described by an isotropic Gaussian distribution. In other words, the components $\{ h_i^{(n)} \in \mathbb{R} \mid i \in [d_H]\}$ of these activation vectors should follow a Gaussian distribution for every activation vector $h^{(n)}$.
>
> Now if the representation space admits priviledged directions, one should *not* expect these components to be Gaussianly distributed. In particular, outlier features will correspond to some of these components to be substantially larger than the average of all components. Therefore, the distribution of activation vectors $\{ h_i^{(n)} \in \mathbb{R} \mid i \in [d_H]\}$ should be heavy-tailed. It turns out that the kurtosis is an ideal metric to measure this heavy-tailedness, since Gaussian distributions are known to have a kurtosis of 3. Therefore, any kurtosis substantially larger than 3 indicates that the components of the activation vector do not follow a Gaussian distribution and that some components are substantially larger than average, which suggests the existence of outlier features.
>
>
> **[1]** Elhage, N., Lasenby, R., & Olah, C. (2023). Privileged bases in the transformer residual stream. Transformer Circuits Thread.
>
> ___
> ## Re Question 1. (Large kurtosis for OpenAI models)
>
> We were also surprised with the extreme kurtosis of OpenAI CLIP models. Since all of these models have similar backbones and are trained with similar hyperparameters, the difference in kurtosis is likely related to the pretraining set used to obtain these models. Since OpenAI never released the full pretraining set of CLIP, we can can only speculate about the possible explanations for this phenomenon.
>
> Our first intuition was that kurtosis increases with the size of the pretraining set. However, a close inspection of *Table 2* is sufficient to dismiss this hypothesis. Indeed, the kurtosis of LAION-2B models is susbtantially smaller than the kurtosis of OpenAI models, in spite of being  pretrained with 2 billion images, which is almost an order of magnitude of the 400 million images used by OpenAI.
>
> A more plausible explanation is the diversity of the pretraining data and could echo some previous work trying to understand the specificities of OpenAI models **[1]**. We have good reasons to think that all the pretraining sets studied in our paper are less diverse than the one used by OpenAI. For the CC-12M and YFCC-15M sets, this boils down to their size that is one order of magnitude bellow the size of OpenAI's pretraining set. Furthermore, YFCC-15M is a subset of the pretraining set used by OpenAI. For the LAION dataset, it boils down to the fact that these datasets are constructed by filtering image-caption pairs scrapped from the web based on the alignment of their representation in CLIP's representation space. This process is likely to filter-out pairs that are substantially different than the ones used by OpenAI. Therefore, it is legitimate to assume that OpenAI's pretraining set is the most diverse dataset discussed in our paper.
>
> The exact interaction between this data diversity and kurtosis would consitute an interesting extension of our work. Unfortunately, it is hampered by the lack of access to OpenAI's pretraining set. A possible workaround would be to gradually reduce the diversity of the LAION datasets with a principled filtering technique and study how this impacts the kurtosis of the resulting models. We leave this interesting avenue for future work.
>
> **[1]** Nguyen, T., Ilharco, G., Wortsman, M., Oh, S., & Schmidt, L. (2022). Quality not quantity: On the interaction between dataset design and robustness of clip. Advances in Neural Information Processing Systems, 35, 21455-21469.

---

> ### Author Response · Authors · 2023-11-18
> **[2/2] Rebuttal to Reviewer YHoY**
>
> ## Re Question 2. (Alternatives to SVD)
>
>
> We thank the reviewer for the interesting suggestion. The reason why we used SVD is to extract directions of the model's representation space $\mathbb{R}^{d_H}$, where $d_H \in \mathbb{N}^+$ is the dimension of the representation space, that are relevant for the computation of the logits by the classification head $W \in \mathbb{R}^{d_Y \times d_H}$, where $d_Y = 1,000$ is the number of ImageNet classes.
>
> If we understood the reviewer's suggestion correctly, the reduced rank regression will consist in stacking the activation $\{ h^{(n)} \in \mathbb{R}^{d_H} \mid n \in [N] \}$, where $N$ is the number of instances for which the activation vectors are computed in a matrix $H \in \mathbb{R}^{d_H \times N}$ in order to find a low rank approximation of $WH \in \mathbb{R}^{d_Y \times N}$, e.g. through a SVD decomposition. We are not sure to understand how this would allow us to distill relevant directions of the model's representation space $\mathbb{R}^{d_H}$, since the matrix $WH$ has an inconsistent shape $d_Y \times N$. If the reviewer is interested in this experiment, could they clarify this? We would be keen to run any additional experiment based on the reviewer's feedback.

---

### Official Review · Reviewer_KW5N · 2023-10-30

**Soundness:** 2 fair
**Presentation:** 2 fair
**Contribution:** 2 fair
**Rating:** 5
**Confidence:** 1

**Summary:**

This paper demonstrates the existence of outlier features and the substantial encoding of multiple concepts in robust models through the study of models like CLIP.

**Strengths:**

This paper provides a systematic investigation of models like CLIP, offering compelling evidence that robust models encode outlier features and a greater variety of concepts. This research is quite interesting.

**Weaknesses:**

1, This paper appears to explain some interesting phenomena but doesn't offer methods for improving model performance. Therefore, I believe the contributions of this research may be relatively limited. As a result, I consider the overall quality of the paper to be at a borderline level.

2, I believe what might be more interesting is understanding why these phenomena occur rather than merely showcasing them.

3, I might have been more eager for the authors to utilize the findings in this paper to inspire some ideas for addressing unresolved problems.

**Questions:**

See Weakness

---

> ### Author Response · Authors · 2023-11-18
> **[1/2] Rebuttal to Reviewer KW5N**
>
> We thank the reviewer for their thoughtful comments and suggestions. Below, we address all the reviewer's remarks. We are keen to address any remaining concerns the reviewer might have during the discussion period.
>
>
> ## Re Weakness 1. and 3. (Impact of findings)
>
> We appreciate the reviewer thoughtful evaluation of our paper. While we understand that proposing an immediate way to leverage our findings would have been ideal, we would like to highlight the nature of our contribution and the broader spirit of scientific exploration.
>
> Our work focuses on identifying a phenomenon within the model's behavior, aiming to deepen the understanding of underlying mechanisms. It is important to note that scientific progress often involves incremental steps, with each study contributing a piece to the larger puzzle. In this context, our findings serve as a foundation for future research and could inspire novel ideas for addressing unresolved problems.
>
> Scientific contributions extend beyond immediate applications and can pave the way for innovative approaches in subsequent studies. By understanding the relationships between effective robustness, outlier features, high kurtosis, and privileged directions, we provide valuable insights that could potentially inform the development of more effective models in the future. We believe that collaboration and building upon prior research are key elements of scientific advancement, and we welcome further exploration and refinement of our work in subsequent investigations.
>
> For example, we see two distinct fields of ML research that could benefit from the phenomena we exhibit in this paper: model quantization and mechanistic interpretability.
>
> **Model Quantization.** Quantization is the process of converting floating point tensor operations of a model into integer tensor operations in order to speed-up inference time and decrease memory consumption. Outlier features are known to occur in LLMs training and make the quantization of LLMs challenging **[1]**. In our paper, we demonstrate that the same type of outlier features occur in the robust zeroshot CLIP models. This implies that the quantization of CLIP models will cause the same challenges as the ones encountered in the LLM litterature. Hence, decreasing the memory consumption and inference time of CLIP models cannot be achieved by using out-of-the-box model quantization.
>
> **Mechanistic Interpretability.** The goal of mechanistic interpretability is to reverse engineer neural networks in order to explain their inner working and achieve higher transparency. The common approach is to analyze each neuron / direction of the model's representation space and to investigate how these neurons form circuits connecting the model's input and output **[2]**. A crucial assumption in mechanistic interpretability is the fact that each neuron activates in the presence of related human concepts, hence allowing us to attach unambiguous meaning to the model's neurons. However, empirical analysis show that neural networks tend to superpose many concepts in some directions of their representation spaces, a phenomenon known as *polysemanticity* **[3]**. This phenomenon is the main obstacle to mechanistic interpretability, without definite solutions at this day. In our paper, we show that robust zeroshot CLIP models tend to superpose more concepts that their nonrobust counterparts (finetuned / supervised models). This implies that the interpretation of robust zeroshot CLIP models through mechanistic interpretability is more challenging.
>
> **[1]** Dettmers, T., Lewis, M., Belkada, Y., & Zettlemoyer, L. (2022). Llm. int8 (): 8-bit matrix multiplication for transformers at scale. arXiv preprint arXiv:2208.07339.
>
> **[2]** Olah, C., Cammarata, N., Schubert, L., Goh, G., Petrov, M., & Carter, S. (2020). Zoom In: An Introduction to Circuits.
>
> **[3]** Elhage, N., Hume, T., Olsson, C., Schiefer, N., Henighan, T., Kravec, S., ... & Olah, C. (2022). Toy models of superposition. arXiv preprint arXiv:2209.10652.

---

> ### Author Response · Authors · 2023-11-18
> **[2/2] Rebuttal to Reviewer KW5N**
>
> ## Re Weakness 2. (Role of signatures in model performances)
>
>
> We fully agree with the reviewer that it would be valuable to understand *why* these signatures emerge when pretraining the model. While our work does not bring a definite answer to this line of research, we argue that it brings two valuable contributions to the field.
>
> **Extension beyond LLMs.** We note that the presence of outlier features and the superposition of many concepts in the model's representation space has been an active area of research in *mechanistic interpretability* (see e.g. **[1]** and **[2]**). While this line of work has established that these phenomena occur in training large language models (LLMs), we believe that our work constitutes a first bridge with vision models through the multimodality of CLIP. Importantly, our work establishes that these signatures are not specific to LLMs and happen when pretraining vision models on large datasets.
>
> **Challenging existing explanations.** Some recent works have tried to explain the emergence of outlier features in LLMs. For instance **[3]** suggested that outlier are created by attention layers in order to encode "no-op" updates in the residual stream. Our work suggests that this explanation is incomplete, as outlier features are also present in CLIP ResNets, without any attention layers.
>
> Interestingly, our signatures are absent from vision models trained on smaller datasets (e.g. ImageNet), as demonstrated in *Sections 3 & 4*. This suggests that the signatures might be a consequence of scaling up the size of the data seen by the model, and are not necessarily tied to a specific modality.
>
> **[1]** Elhage, Nelson, Robert Lasenby, and Christopher Olah. "Privileged bases in the transformer residual stream." Transformer Circuits Thread (2023).
>
> **[2]** Bricken, Trenton, et al. "Towards Monosemanticity: Decomposing Language Models With Dictionary Learning." Transformer Circuits Thread (2023).
>
> **[3]** Bondarenko, Yelysei, Markus Nagel, and Tijmen Blankevoort. "Quantizable Transformers: Removing Outliers by Helping Attention Heads Do Nothing." arXiv preprint arXiv:2306.12929 (2023).

---

> ### Author Response · Authors · 2023-11-21
> **Follow-up to the Rebuttal for Reviewer KW5N**
>
> We again thank the reviewer for their useful comments. We hope that our rebuttal has addressed any remaining concern about the paper.  If not, we would like to kindly ask the reviewer to engage in discussion before the end of the discussion period (tomorrow). Otherwise, we hope that the reviewer will consider updating their recommendation accordingly.

---

### Official Review · Reviewer_qBSE · 2023-11-01

**Soundness:** 3 good
**Presentation:** 3 good
**Contribution:** 2 fair
**Rating:** 6
**Confidence:** 3

**Summary:**

The paper studies the features learned by robust multimodal models. This paper explores the concept of robustness in multimodal models, specifically looking at the differences between robust and non-robust models and uncovering two signatures of robustness in the representation spaces of these models. The authors find that robust models have outlier features, which are highly activated components of the representation space. These outlier features induce privileged directions in the representation space, which are important for the model's performance. The authors also find that robust models encode more unique concepts than less robust models. This leads to polysemy, where a single representation can be used to represent multiple concepts. Additionally, they demonstrate that privileged directions in the model's representation space explain the model's predictive power. The paper analyzes multiple robust multimodal models trained on various pretraining sets and backbones. Overall, this paper provides valuable insights into the nature of robustness in multimodal models and sheds light on the factors that contribute to their success.

**Strengths:**

1. This paper makes a significant contribution to the field of multimodal models by uncovering two signatures of robustness in the representation spaces of these models. This is a novel approach that has not been explored in previous research. It provides a new understanding of the features that make robust multimodal models robust. The authors identify two key features: outlier features and privileged directions. Outlier features are highly activated components of the representation space, while privileged directions are directions that are important for the model's performance. The authors show that both of these features are more prevalent in robust models than in less robust models.

2. The authors analyze different robust multimodal models trained on various pretraining sets and backbones, providing a comprehensive and rigorous analysis of the factors that contribute to robustness in these models. The paper uses a variety of methods to validate its findings. In addition to using activation kurtosis and singular value decomposition (SVD) to identify outlier features and privileged directions, the authors also use concept probing to show that robust models encode more unique concepts. This provides strong evidence that the authors' findings are not just artifacts of the specific methods they used.

3. The findings of this paper have practical implications for the development of robust multimodal models. The authors demonstrate that outlier features and privileged directions in the model's representation space are key factors in the model's success, which can inform the development of more robust models in the future. The authors' identification of outlier features and privileged directions suggests that these features should be preserved in model training. This could be done by using training objectives that encourage the model to learn these features, or by using regularization techniques to prevent the model from overfitting to the training data.

**Weaknesses:**

1. The paper only analyzes CLIP robust multimodal models with different backbones, which may not be representative of all possible models. This limits the generalizability of the findings. There are many other large scale multimodal models, which should be included, i.e., BLIP [1], FLAVA [2].

2. The paper only focuses on robust models and does not compare them to non-robust models. This makes it difficult to determine the extent to which the findings are specific to robust models. See more questions in Questions section.

3. Lack of explanation of outlier features: While the paper identifies outlier features as a key factor in robustness, it does not provide a clear explanation of what these features are or how they contribute to robustness. While the paper does discuss the practical implications of the findings, it could have gone into more detail about how these findings can be applied in practice to develop more robust multimodal models, i.e., VK-OOD [3]

[1]Li, J., Li, D., Xiong, C., & Hoi, S. (2022, June). Blip: Bootstrapping language-image pre-training for unified vision-language understanding and generation. In International Conference on Machine Learning (pp. 12888-12900). PMLR.

[2]Singh, A., Hu, R., Goswami, V., Couairon, G., Galuba, W., Rohrbach, M., & Kiela, D. (2022). Flava: A foundational language and vision alignment model. In Proceedings of the IEEE/CVF Conference on Computer Vision and Pattern Recognition (pp. 15638-15650).

[3]Wang, Z., Medya, S., & Ravi, S. N. (2023). Differentiable Outlier Detection Enable Robust Deep Multimodal Analysis. arXiv preprint arXiv:2302.05608.

**Questions:**

1. What are the two signatures of robustness in the representation spaces of multimodal models, and how do they contribute to the models' success?

2. How do outlier features in robust multimodal models differ from those in non-robust models, and what is their role in robustness?

3. What are privileged directions in the model's representation space, and how do they explain the model's predictive power?

4. In Figure 1, what are baseline models and baseline models fit training on? Directly from ImageNet? How can this accuracy compare with the multimodal fine-tuning ones?

5. Figure 2 is a little bit hard to read and understand. The authors should present and well-explain the figures clearly.

6. Why only use kurtosis value to determine outlier features? The model of reference work has different number of parameters, how did authors choose the same one as their work?

---

> ### Author Response · Authors · 2023-11-18
> **[1/3] Rebuttal to Reviewer qBSE**
>
> We thank the reviewer for their thoughtful comments and suggestions. Below, we address all the reviewer's remarks. We are keen to address any remaining concerns the reviewer might have during the discussion period.
>
> ## Re Weakness 1. (Restriction to CLIP)
>
>  We have taken the reviewers recommendation into account and confirmed our findings on multimodal models outside of the CLIP family, namely CoCa models **[1]**. Beyond that, we have futher extended the experimental breadth of our paper by extending our analysis to larger scale models with ViT-L-14 backbones. All these results have been added to *Appendix C.1* of the updated manuscript.
>
> Below, we verify that the empirical results from our paper extend to these models. First we confirm that all these models have high effective robustness when used as zero-shot classifiers. As in the paper, we observe that finetuning on ImageNet decreases the effective robustness of these classifiers.
>
> |**Backbone**|**Pretraining Data**|**Zero-shot ER**|**Finetuned ER**|
> |---|---|---|---|
> |COCA ViT-B-32|LAION-2B|25%|14%|
> |COCA ViT-L-14|LAION-2B|34%|21%|
> |CLIP ViT-L-14|OpenAI| 37% | 21%|
> |CLIP ViT-L-14|LAION-400M|32%|20%|
> |CLIP ViT-L-14|LAION-2B|32%|21%|
> |CLIP ViT-L-14|DataComp|37%|24%|
> ||
>
> Next, we show that all the zero-shot models have high kurtosis, which implies the existence of outlier features in their representation space. Additionally, we show that finetuning again decreases the kurtosis.
>
> |**Backbone**|**Pretraining Data**|**Zero-shot kurtosis**|**Finetuned kurtosis**|
> |---|---|---|---|
> |COCA ViT-B-32|LAION-2B|12.0|3.6|
> |COCA ViT-L-14|LAION-2B|15.5|4.6|
> |CLIP ViT-L-14|OpenAI| 60.8 | 4.6|
> |CLIP ViT-L-14|LAION-400M|20.3|5.2|
> |CLIP ViT-L-14|LAION-2B|66.2|6.9|
> |CLIP ViT-L-14|DataComp|37.4|4.6|
>
>
> Finally, we check that zero-shot model encodes more concepts. Again, we see that finetuning removes some concepts from the model's representation space.
> |**Backbone**|**Pretraining Data**|**Zero-shot #concept**|**Finetuned #concept**|
> |---|---|---|---|
> |COCA ViT-B-32|LAION-2B|674|530|
> |COCA ViT-L-14|LAION-2B|747|629|
> |CLIP ViT-L-14|OpenAI| 704 | 623|
> |CLIP ViT-L-14|LAION-400M|683|613|
> |CLIP ViT-L-14|LAION-2B|704|633|
> |CLIP ViT-L-14|DataComp|684|619|
>
>
> In addition to these multimodal models, we have conducted a similar analysis on a robust SimCLR ViT-B-16 trained on the YFCC-15M dataset. This model is an interesting case study, as it is robust without being multimodal. Interestingly, our analysis in *Appendix G* shows that this model also has privileged directions and a large number of concepts encoded in its representation space. Please refer to *Appendix G* for more details.
>
> Finally, we would like to discuss the BLIP model suggested by the reviewer. We note that this model is implicitly studied in our paper, as it is endowed with the ViT-B-16 pretrained by OpenAI. Hence, we deduce that the analysis in our paper extends to BLIP.
>
> **[1]** Jiahui Yu, Zirui Wang, Vijay Vasudevan, Legg Yeung, Mojtaba Seyedhosseini, Yonghui Wu. CoCa: Contrastive Captioners are Image-Text Foundation Models
> ___
> ## Re Weakness 2. + Question 2. (Lack of comparison to non-robust models)
>
> We would like to clarify that the analysis in our paper *does* incorporate comparisons with non-robust models. Indeed, by looking at *Table 1*, we observe that ImageNet supervised models have no (or negligeable) effective robustness, making them non-robust models. Interestingly, the finetuned models have a medium effective robustness. By combining zero-shot, finetuned and supervised models for each backbone, we then have models with high, medium and no effective robustness.
>
>
> Each section of our paper covers all 3 model types, as can be observed in e.g. *Table 2*, *Figure 2*, *Table 3* and *Figure 4*. In particular, we discover the 2 following robustness signatures:
>
> 1. High kurtosis inducing priviledged directions in the model's representation space.
> 2. A high number of concepts encoded in the model's representation space.
>
> Our experiments demonstrate that these signatures distinguish robust models from non-robust ones.
>
> In particular, the answer to the reviewer's Question 2 can be found in Table 2: Outlier features in robust models differ from those in non-robust models in that they exist only for robust models (activation kurtosis of $\gg$ 3), but not for non-robust models (activation kurtosis of $\approx$ 3).
>
>
> Similarly, we can deduce the robustness signature nature of the high number of encoded concepts in the model's representation space from *Table 3*: for each backbone, the amount of encoded concept is much higher for the robust zero-shot models than for the other, less robust models.
>
> These observations reinforce the idea that these properties indeed distinguish robust models from the rest, and hence constitute signatures.

---

> > ### Comment · Reviewer_qBSE · 2023-11-22
> >
> > Thanks for Thanks the authors for the responses. Several concerns are explained. I think it is better to clearly explained how to define three types of robustness in the paper. But why use ImageNet supervised models as no effective robustness? Is it the models trained on ImageNet, if so, It is not multimodal models.

---

> ### Author Response · Authors · 2023-11-18
> **[2/3] Rebuttal to Reviewer qBSE**
>
> ## [1/2] Re Weakness 3. + Questions 3., 5., and 6. (Details on outlier features & priviledged directions)
>
> We appreciate the opportunity to elaborate on a crucial point: the connection between privileged directions, outlier features, high kurtosis.
>
> Privileged directions are one of our contributions and denote outlier features that not only receive substantially higher activations but also contribute significantly to the logits.
>
> Outlier features are directions that receive a substantially higher activation, this term was introduced in  **[1]**. Compared to Privileged Direction, outliers features may or may not contribute significantly to the logits.
>
> A high kurtosis signals that some directions of the model’s representation space receive a substantially higher activation than the other directions. However, since the kurtosis metric defined in *Equation (2)* aggregates over all components of the activation vectors, it is not possible to identifies which directions of the representation space are outlier features.
>
> **Summary**. In summary, high kurtosis indicates the presence of outlier features, without identifying them. Outlier features represent directions with high activations, regardless of their contribution to the logits. Privileged directions are a subset of outlier features that also contribute substantially to the logits and consequently, to the final predictions.
>
>
> Below we offer a more formal explaination where we also explain how we identify priviledged directions using the importance score.
>
>
> **Large activations in some directions.** Consider a basis $\{ v_i \in \mathbb{R}^{d_H} \mid i \in [d_H] \}$ of the representation space $\mathbb{R}^{d_H}$, where $d_H \in \mathbb{N}^+$ is the dimension of the representation space. Let us assume that the direction $v_j$ receive substantially higher activations, hence corresponding to an outlier feature. This implies that the absolute cosine similarity of an activation vector $h^{(n)} \in \mathbb{R}^{d_H}$ with this direction will typically be higher than average: $|\cos (v_j, h^{(n)})| \gg \frac{1}{d_H} \sum_{i=1}^{d_H} |\cos (v_i, h^{(n)})|$. This corresponds to the second factor in our importance score $\mathrm{Importance}(j)$ defined in *Equation (4)*.
>
> **Contribution to logits.** Now what guarantees that this outlier feature $v_j$ is indeed important for the model? As we argue in the paper, this direction of the latent space needs to play a role in the computation of the logits. In order to restrict to directions of the latent space that matter for the classifier, we study the directions of the latent space corresponding to right singular vectors $\{ v_i \in \mathbb{R}^{d_H} \mid i \in [\mathrm{rank}(W)] \}$ of the classification head $W \in \mathbb{R}^{d_Y \times d_H}$, where $d_Y = 1,000$ is the number of ImageNet classes. Indeed, this set of vector spans the subspace of $\mathbb{R}^{d_H}$ that is orthogonal to the kernel of the classification head $\mathrm{span} \{ v_i \in \mathbb{R}^{d_H} \mid i \in [\mathrm{rank}(W)] \} = \ker(W)^{\perp}$, which means that they allow us to decompose any activation vector $h \in \mathbb{R}^{d_H}$ whose contribution to the logits is nonzero. Now the outlier feature $v_j$ has a substantial contribution to the logit only if it corresponds to a large singular value $\sigma_j \gg 0$. This motivates our characterization of *priviledged directions* as directions of the representation space that receive a large activation (and, hence, correspond to outlier features) *and* that have a substantial contributions to the logits. This extra contribution corresponds to the first factor in our importance score $\mathrm{Importance}(j) \in \mathbb{R}^+$ defined in *Equation (4)*. Priviledged directions are then characterized by an importance score substantially higher than average $\mathrm{Importance}(j) \gg \frac{1}{d_H} \sum_{i=1}^{\mathrm{rank} (W)} \mathrm{Importance}(i)$.
>
> **Interpretation of Figure 2.** This leads us naturally to *Figure 2* that shows how the importance scores $\{ \mathrm{Importance}(i) \mid i \in [\mathrm{rank}(W)] \}$ are distributed for various models. Each point on this plot corresponds to a direction of the model's representation space. As we can see, zero-shot models have priviledged directions having significantly higher importance (corresponding blue points that are substantially above the bulk). finetuned models still have priviledged directions (corresponding blue points that are above the bulk), although their gap of importance with the bulk is smaller. Nonrobust supervised models have no priviledged directions, as all their importance scores are comparable (no green point stands out from the bulk).
>
> **[1]** Dettmers, T., Lewis, M., Belkada, Y., & Zettlemoyer, L. (2022). Llm. int8 (): 8-bit matrix multiplication for transformers at scale. arXiv preprint arXiv:2208.07339.

---

> > ### Author Response · Authors · 2023-11-18
> > **[3/3] Rebuttal to Reviewer qBSE**
> >
> > ## [2/2] Re Weakness 3. + Questions 3., 5., and 6. (Details on outlier features & priviledged directions)
> >
> > **Priviledged directions interpretation.** Finally, we would like to emphasize that the interpretation of outlier features is discussed in paragraph *Interpreting privileged directions* of Section 4. In particular, we show that the priviledged directions encode generic concepts describing textures with alternating patterns (e.g. the perforated concept). A comparison with the privileged directions of the supervised models demonstrates that finetuning replaces this generic concepts with more concrete concepts (e.g. the tennis court concept). We recall that the nonrobust supervised models have no outlier features, as demonstrated by *Figure 2*.
> >
> > ___
> > ## Re Question 1. (Role of signatures in model performances)
> >
> > Recall that the two robustness signatures are
> > 1. High kurtosis inducing priviledged directions in the model's representation space.
> > 2. A high number of concepts encoded in the model's representation space.
> >
> > We fully agree with the reviewer that it would be valuable to understand how our 2 signatures of robustness contribute to the model's performance. We believe that answering this question subsumes a good understanding of *why* these signatures emerge when pretraining the model. While our work does not bring a definite answer to this line of research, we argue it advances the understanding about robustness with two valuable contributions:
> >
> > **Extension beyond LLMs.** We note that the presence of outlier features and the superposition of many concepts in the model's representation space has been an active area of research in *mechanistic interpretability* (see e.g. **[1]** and **[2]**). While this line of work has established that these phenomena occur in training large language models (LLMs), we believe that our work constitutes a first bridge with vision models through the multimodality of CLIP. Importantly, our work establishes that these signatures are not specific to LLMs and happen when pretraining vision models on large datasets.
> >
> > **Challenging existing explanations.** Some recent works have tried to explain the emergence of outlier features in LLMs. For instance **[3]** suggested that outlier are created by attention layers in order to encode "no-op" updates in the residual stream. Our work suggests that this explanation is incomplete, as outlier features are also present in CLIP ResNets, without any attention layers.
> >
> > Interestingly, our signatures are absent from vision models trained on smaller datasets (e.g. ImageNet), as demonstrated in *Sections 3 & 4*. This suggests that the signatures might be a consequence of scaling up the diversity of the data seen by the model, and are not necessarily tied to a specific modality.
> >
> > **[1]** Elhage, Nelson, Robert Lasenby, and Christopher Olah. "Privileged bases in the transformer residual stream." Transformer Circuits Thread (2023).
> >
> > **[2]** Bricken, Trenton, et al. "Towards Monosemanticity: Decomposing Language Models With Dictionary Learning." Transformer Circuits Thread (2023).
> >
> > **[3]** Bondarenko, Yelysei, Markus Nagel, and Tijmen Blankevoort. "Quantizable Transformers: Removing Outliers by Helping Attention Heads Do Nothing." arXiv preprint arXiv:2306.12929 (2023).
> > ___
> > ## Re Question 4. (Baseline models in ER)
> >
> > We confirm that all the baseline models in *Figure 1* are trained on ImageNet only. They mostly differ by the backbone on which they rely, their training objectives, their hyperparameters, and the type of augmentation used to train the models. These models and their performance have been collected by **[1]**, where the complete list can be found.
> >
> > Following-up on the reviewer's question to why the accuracy of some of these models are higher than finetuned CLIP models (if we understand correctly), we note that none of the finetuned CLIP models are state of the art. State of the art performance on ImageNet is only achieved by CLIP models at larger scale (1B+ parameters) than experimented in this paper. In the smaller model regime explored in this paper, methods that train only on ImageNet such as Masked Autoencoders **[2]** lead to the best performance. However, training on ImageNet only leads to models with far lower effective robustness.
> >
> > **[1]** Rohan Taori, Achal Dave, Vaishaal Shankar, Nicholas Carlini, Benjamin Recht, and Ludwig Schmidt. Measuring robustness to natural distribution shifts in image classification. Advances in Neural Information Processing Systems, 33:18583–18599, 2020.
> >
> > **[2]** He, Kaiming, et al. "Masked autoencoders are scalable vision learners." Proceedings of the IEEE/CVF conference on computer vision and pattern recognition. 2022.

---

> ### Author Response · Authors · 2023-11-21
> **Follow-up to the Rebuttal for Reviewer qBSE**
>
> We again thank the reviewer for their useful comments. We hope that our rebuttal has addressed any remaining concern about the paper. If not, we would like to kindly ask the reviewer to engage in discussion before the end of the discussion period (tomorrow). Otherwise, we hope that the reviewer will consider updating their recommendation accordingly.

---

> ### Author Response · Authors · 2023-11-22
>
> We are pleased that the reviewer finds several of their concerns addressed and we thank the reviewer for engaging in the discussion on their remaining concern, which we address below.
>
> While the non-robust models we compare to are not multimodal (since they were trained in supervised way on ImageNet), we believe that these are the best non-robust models we could compare to. This is for the following reasons:
> - The only non-robust multimodal models the authors are aware of are the ImageNet-Captions CLIP models from **[1]**. However, none of these models achieves more than 35% accuracy on ImageNet (see green dots in Figure 1 in **[1]**), that is about half the accuracy of most of the robust models that we investigate in our paper (in other contexts a model with such performance on ImageNet would be called degenerate).
> - On the other hand, the non-robust supervised models that we do compare to in our paper achieve ImageNet accuracies that are comparable to most of the robust models that we analyze. They also share the exact same backbone with the robust multimodal models.
>
> For these reasons, we believe that the non-robust supervised models we compare to in the paper are currently the best possible comparison.
>
> We hope this answer also addresses the reviewer’s last concern. We would like to ask the reviewer if this is the case, and if there is any other concern that the reviewer believes should be clarified or changed for this paper to be accepted.
>
>
> **[1]** Alex Fang, Gabriel Ilharco, Mitchell Wortsman, Yuhao Wan, Vaishaal Shankar, Achal Dave, Ludwig Schmidt. Data Determines Distributional Robustness in Contrastive Language Image Pre-training (CLIP).

---

> > ### Comment · Reviewer_qBSE · 2023-11-22
> >
> > Thanks for the further explanations, I raise my rating to marginally above.

---

> ### Author Response · Authors · 2023-11-23
>
> We thank again the reviewer for their engagement and for their valuable feedback.

---

### Author Response · Authors · 2023-11-18
**Rebuttal summary**

We thank all the reviewers for their useful comments and suggestions. We would like to highlight 3 ways in which the reviewers helped us reinforce our work with the rebuttal.

1. We have extended the experimental breadth of our work and validated our results both on larger scale backbones (ViT-L-14) and beyond the CLIP-family of models (CoCa models). We now have a total number of 18 models analyzed.
2. We clarified the link that exists between outlier features, activation kurtosis and priviledged directions.
3. We better anchored our findings in the relevant literature through connections with low-rank approximation of neural networks, model quantization and mechanistic interpretability.

All of these points, together with more specific queries, are addressed in our rebuttal. We are keen to discuss all of these during the rest of the discussion period.

---

### Meta-Review · Area_Chair_UTN7 · 2023-12-11

**Metareview:**

This paper delves into the feature disparity in the last layer between robust and non-robust models. Their primary findings indicate that: 1) robust multimodal models possess outlier features, and 2) features within robust models encode a greater variety of concepts.

While the authors have adeptly addressed some concerns raised by reviewers during the rebuttal stage, several still persist. For example, as pointed out by Reviewer 8h4M, the ideas would benefit from being further developed, e.g., investigating if some of the observations that this paper makes could be interpreted in terms of the linear dimension (that is, the lowest dimensional subspace that can capture variation in the data) if one combines all the privileged directions into a subspace. Another statistic that might be interesting to look at is stable rank.

In addition, one recent work at NeurIPS-23 [1] revealed that the better effective robustness of CLIP models comes from incorrect evaluation using imprecise ID test distribution. As a result, the effective robustness of CLIP greatly diminishes using their evaluation. Consequently, the findings presented in this work might no longer be impactful, as it primarily concentrates on effective robustness.

Lastly, this work mainly focuses on discovering the difference between robust and non-robust models without developing a deeper understanding of why this happens and how to use these observations to further improve robustness. This greatly limits the contribution of this work.

[1] Shi. et al,  Effective Robustness against Natural Distribution Shifts for Models with Different Training Data

**Justification For Why Not Higher Score:**

1) This work mainly focuses on discovering the feature difference between robust and non-robust features without developing deeper insights into their observations.

2) Without inspiring insights, the work does not propose methods to enhance robustness based on their observations.

3) The discovered observations build upon an imprecise condition that CLIP models significantly improve effective robustness.

**Justification For Why Not Lower Score:**

N/A

---

### Decision · Program_Chairs · 2024-01-16

Reject